# Collaborative Mobile-Learning Architecture Based on Mobile Agents

**Samer Atawneh [1] , Mousa Al-Akhras [1,2,\*] , Iman AlMomani [2,3] , Anas Liswi [4] and Mohammed Alawairdhi [1]**

1   College of Computing and Informatics, Saudi Electronic University, Riyadh 11673, Saudi Arabia;
    satawneh@seu.edu.sa (S.A.); malawairdhi@seu.edu.sa (M.A.)
2   King Abdullah II School of Information Technology, The University of Jordan, Amman 11942, Jordan;
    i.momani@ju.edu.jo or imomani@psu.edu.sa
3   Department of Computer of Science, Collage of Computer and Information Sciences, Prince Sultan
    University, Riyadh 11586, Saudi Arabia
4   Amman Stock Exchange, Amman 11121, Jordan; aliswi@ase.com.jo
\*   Correspondence: m.akhras@seu.edu.sa or mousa.akhras@ju.edu.jo; Tel.: +96-6112613500

**Abstract:** The connection between collaborative learning and the new mobile technology has become tighter. Mobile learning enhances collaborative learning as learners can access information and learning materials from anywhere and at any time. However, supporting efficient mobile learning in education is a critical challenge. In addition, incorporating technological and educational components becomes a new, complex dimension. In this paper, an efficient collaborative mobile-learning architecture based on mobile agents is proposed to enhance learning activity and to allow teachers and students to collaborate in knowledge and information transfer. A mobile agent can control its own actions, is able to communicate with other agents, and adapts in accordance with previous experience. The proposed model consists of four components: the learner agent, the teacher agent, the device agent and the social agent. The social agent plays the main role in the collaborative tasks since it is responsible for evaluating the collaborative interactions among different learners. Additionally, it offers an evaluation indicator for the learners' collaboration and supplies the teacher with learner's collaboration reports. The proposed model is evaluated by introducing a collaborative mobile-learning case study applied to two full classes of undergraduate students. To conduct the model experiments, students were asked to complete a questionnaire after they used the proposed model. The questionnaire results statistically revealed that the proposed architecture is easy to use and access, well-organized, convenient, and facilitates the learning process. The students thought the proposed m-learning application should complement rather than replace the traditional lectures. Moreover, the experimental results show that the proposed collaborative mobile learning model enhances the learner's skills in problem solving, increases the learner's knowledge in comparison with individual learning, and social agent encourages learners for more participation in the learning tasks. Based on the experiments conducted, the authors found that the proposed model can improve the quality of the learning process by assessing learners' and groups' collaboration, and it can help teachers make learners improve how they work in groups. This also provides various ways of assessing learners abilities and skills in groups. It is also possible to integrate the collaborative e-learning with the proposed collaborative m-learning.

**Keywords:** mobile learning; m-learning; collaborative learning; improving classroom teaching; mobile agents; case study

## 1. Introduction

Technology plays an increasingly important role in education to enhance learning and teaching activities. Rodríguez et al. [1] confirmed the fact that information technology (IT) tools, such as mobile devices, have multiplied the possibilities to carry out collaborative work. Mobile learning (m-learning) is a hot research area that enables learners to access learning resources ubiquitously from anywhere and at any time using their mobile devices [2,3]. Most of the progress made in this field has been influenced by the evolving mobile technological infrastructure [4] and the development of wireless technology [5]. M-learning takes advantage of mobile wireless communication where the learners are no longer restricted to time, space or cable network infrastructure. Learners can learn flexibly, conveniently and at anytime, anywhere for different purposes, in different ways. Additionally, m-learning is considered real-time learning as it can help collaborative learners find a solution for the encountered problems through a mobile device and, therefore, m-learning can provide opportunities for the learner to interact with other learners and to have the possibility of giving feedback. This learning process should help learners to improve their grades, confidence, communication ability, knowledge sharing, student interaction and learning efficiency [6]. The United Nations Educational, Scientific and Cultural Organization (UNESCO) recommended adopting technology to ensure access to mobile devices to allow students to access learning possibilities [7]. The organization has asserted that m-learning enhances the students' results and has a great potential in improving the quality of learning process [7]. In addition, m-learning has undergone major leaps and collaborative learning has been enhanced by applying mobile technology [8].

In collaborative learning, learners acquire and build their knowledge by interacting with other learners within a group [1,9]. M-learning is a helpful tool to accelerate the learning process and it encourages both collaborative and independent learning experiences [10]. It enhances collaborative learning by providing new opportunities for learners to interact using mobile devices. This encourages learners to be more active in the learning process [1,11]. In addition, collaborative learning using technology is a very useful approach and accepted as a basic learning form, and is widely adopted [12], and m-learning models can provide an efficient learning environment that enhances the learning process with a collaborative learning between learners and teachers [13]. A recent study confirmed the positive impact of collaborative m-learning on learners, in both their class participation and learning motivation [14]. It also revealed that collaborative m-learning leads to active engagement and improved class performance.

However, despite the fact that there is a lack of m-learning models [13], the main challenge is how to provide an efficient m-learning environment that enhances the learning process with collaborative learning between learners and teachers. Therefore, the aim of this research was to design and develop an efficient architectural model for a collaborative m-learning environment by effectively adopting mobile agents to enable learners to access learning resources ubiquitously from anywhere and at any time using mobile devices. A mobile agent is a composition of software and data that is able to move from one host to another and continue its execution at the destination. A mobile agent works independently and can control its own actions, is able to communicate with other agents, and adapts in accordance with previous experience. The mobile agent has the ability to move in different mobile device environments with different capabilities and limitations. Another benefit of mobile agents is a study-ability, which refers to the ability of the agent to adjust its own actions according to the information it gathers from its surroundings communication. Moreover, mobile agents can reduce the network load by allowing users to package a conversation and dispatch it to a destination host where interactions take place locally. This the m-learning system with great benefits due to wireless communication limitations.

The proposed model design consists of four components: the learner agent, the teacher agent, the device agent and the social agent. The learner agent observes the learner activity and updates the learner profile (achievement, understanding, and performance) to enhance the learning development process. The teacher agent helps teachers to guide the learners' study, and analyzes the learner's

information to give an informative feedback. The teacher agent decides which learners understand the learning tasks and gives clarifications. It can guide the learner for learning difficulties of his/her study.

The social agent plays the main role in the collaborative tasks since it is responsible for continuously evaluating the collaborative interactions among different learners and monitoring the group collaboration that includes managing group interactions, monitoring collaborative learning activities, evaluating group performance, and increasing collaborative learning within the group. It also increases the learner's social contribution in discussion, counter-suggestion, evaluation, and elaboration. In collaborative learning, a participant's contribution is considered as part of a group's contribution.

The main contributions of this research are:

1.　Building an m-learning model design based on agents.
2.　Building a collaborative mobile-learning model design based on agents.
3.　Building collaborative mobile learning based on a social agent.
4.　Implementing a collaborative mobile-learning model based on agents and presenting a case study.

The significance of the proposed m-learning model lies in allowing students to take a control of their own learning and to create a more robust collaboration mobile-learning environment. In this research, the following research methodology steps are followed: (1) conduct a comprehensive survey in m-learning, mobile agent, adaptive content, and collaboration learning; (2) construct a working scenario using a specific architecture; (3) define a specific architecture with the main players and the interaction among different components; (4) define the use cases for the working scenario; (5) run a simulation and develop a working system for the proposed architecture; (6) deploy the system and study users' satisfaction; and, finally, (7) analyze and study the results in order to magnify the enhancements and the advantages of the proposed approach.

The rest of this paper is organized as follows: Section 2 reviews the literature and presents the concepts of educational paradigms, collaborative learning, mobile agents' services and security, and collaborative learning models. A design for an m-learning model based on agents as a basic model is proposed in Section 3. Section 4 proposes a design for a collaborative m-learning model based on agents and describes the main components of the model. The architectural model analysis and social agent tasks in a collaborative m-learning environment are presented in Section 5. In Section 6, the experiments and simulation of the proposed architectural model are discussed. A case study is presented in Section 7. Finally, in Section 8 concluding marks are made and some possibilities of future works are presented.

## 2. Background and Literature Review

This section discusses current educational paradigms, collaborative learning, mobile agents and their services and security, and collaborative learning models.

### 2.1. Educational Paradigms

Education is the process by which society deliberately transmits its accumulated knowledge, skills, and values from one generation to another [15]. Learners experience various educational methods during their study. These methods of education differ mainly in the way they organize the relation between teachers and the learners. According to Martel [16], these methods are classified into three situations: (1) a frontal situation, where participants have individual activities with no relations with other participants, the teacher coordinates controls and has a central position in the exchange and in communication; (2) an open situation, where participants can cooperate freely with their peers or with the teacher(s). In this situation, each participant is supposed to produce results; and (3) a collective situation, where participants cooperate in order to solve a collective problem which has been given to them. In this situation, any participant's contribution is considered as part of a group's contribution. These situations can be combined to produce an integrated educational situation.

The traditional learning approach in education and training fields, which was based on face-to-face learning, has evolved into distance learning (D-learning), where learners are separated from the teachers [17]. D-learning can take many shapes and it has evolved into electronic learning (E-learning), i.e., learning through the Internet or over an intranet [18], and finally to Mobile Learning (m-learning). The main concern in E-Learning is the delivery of content and interaction via all electronic media, including Internet, intranets, satellite broadcast, audio/video tape, interactive TV, and CD-ROM. M-learning means learners can access learning resources and communicate or cooperate with other learners from anywhere and at any time using mobile devices.

Ally [19] describes a Framework for the Rational Analysis of Mobile Education (FRAME). This model shows that mobile learning is composed of the intersection between three parts: mobile technologies, human learning capacities, and social interaction [13]. The FRAME model (illustrated in Figure 1) represents the three fundamental intersected components: device (D), learner (L), and social interaction (S). The interaction between two circles creates an attribute belong to both aspects. D and L create device usability (DL), L and S create an interaction learning (LS), D and S create a social technology (DS). The intersection of the three aspects creates mobile learning (DLS). DLS provides the collaboration among learners, the ability to access the information. It considered as a mediation that includes the interaction between learner, communities and information.

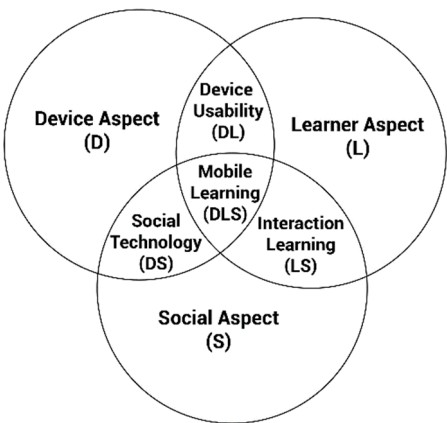

**Figure 1.** The Framework for the Rational Analysis of Mobile Education (FRAME) Model.

## 2.2. Collaborative Learning

Collaborative learning helps developing teamwork skills and facilitates the acquisition of specific knowledge [20]. It has been pointed out that collaborative learning produces intellectual synergy of many minds brought to bear on a problem, and it often leads to better understanding on the part of learners [21]. The term "collaborative learners" in the educational field refers to studying course material or sharing course assignments (problem solving), so a student acquires and builds his/her knowledge by interacting with others within a group [9].

Dillenbourg [9] provides general concerns for developing ways to increase the probability of the occurring interactions among people to have collaborative learning. The main phases of this collaborative learning are: (1) set up initial conditions, which includes design the situation; (2) specify the collaboration contract with a scenario based on roles by defining a clear specification of roles for each learner; (3) support learner's interactions by providing an interface where users communicate with a set of pre-defined buttons; and finally (4) monitor and regulate the interactions.

Deejring [22] proposed and designed a web-based learning model using collaborative learning techniques to enable students to construct knowledge and, therefore, enhance students' competency in higher education. Recently, Al-Rahmi and Zeki [23] proposed a model of using social media for collaborative learning. Their study reported significant impacts (direct and indirect) of several factors on collaborative learning which might lead to enhanced students' learning performance. The study

also revealed that the high satisfaction by students using social media enhances the collaborative learning that leads to a better learning performance.

### 2.2.1. Collaboration and E-Learning

Computer Supported Collaborative Learning (CSCL) supports collaborative learning using technologies to promote students' interaction and collaboration, and maximize their learning achievement [24]. This technology allows learners who are far apart to collaborate online. For example, teachers may provide a blog to the students in their classroom with links to web sites, which includes learning content or research tasks. Teachers may have students use blogs as learning reflections, story writing, etc. Viewers can leave comments, which aid writers in their writing development. Also there are various web-based applications that allow groups of students to work together on common documents in various formats either synchronously or asynchronously [25]. Instructors are considered a central role in enhancing online collaborative learning. Shank [25] shows that strong instructor supports, frequent instructor–student interaction, and advanced organizational skills are important elements of successful online collaborative learning.

### 2.2.2. Collaboration and M-Learning

Mobile Computer Supported Collaborative Learning (MCSCL) refers to the practice of using mobile computing as a way of encouraging learners' interaction in the context of joint activity [26]. M-learning offers new opportunities for students to collaborate in learning. Such opportunities enhance the learner interaction where the use of mobile devices allows creating a learning situation that is hard to achieve using other technologies. When collaboration is needed, the interaction occurs between learners using their mobile devices to allow them to be more active and engaged in the learning process. MCSCL helps to organize and manage the learning information, to allow a negotiation among learners, and to encourage and coordinate the learning activity [11]. Mobile devices operate in different ways and have different capabilities; therefore, some learning resources may not be in a format that is acceptable for different learners' needs and that fits the capabilities of different mobile devices, consequently content adaptation is needed to provide learners with an appropriate course view [4,27].

### 2.3. Mobile Agent

This section highlights the advantages and disadvantages of existing platforms in a mobile environment and the challenges that are normally found in the current mobile applications based on agents. In addition, the good aspects of mobile agents that motivated the authors to use them in the m-learning process are presented.

A mobile agent is a software component that runs on an execution environment (traditionally known as a place) and can move from a place to another within the same host or between different hosts [28]. It can sense the changes in the environment and acts according to those changes. In addition, a mobile agent has control over its own actions, is able to communicate with other agents, and adapts in accordance with previous experience [29]. Mobile agent technology for a wireless environment has many advantages that make it a good solution for many areas. All existing platforms provide many services to mobile agents but they differ in some aspects like performance, scalability, and communication. On the other hand, existing platforms share the main aspects, which are execution, communication, mobility, tracking, directory, and security.

Many security issues arise when using a mobile agent on wireless environment and, therefore, it is important to ensure the privacy and integrity of data, and consider authentication and trust issues. As data are transmitted on wireless communication over mobile devices, data will not be received only by the destination but also by any node within the range of the communications device. Therefore, most wireless communications protocols (such as Wi-Fi or Bluetooth) encrypt the data before transmission. However, in some cases the communications can be unencrypted (such as public access point). The encryption is an important feature for the transfer of a mobile agent on wireless

environment to avoid exposing its code and data and, therefore, a mobile agent should be able to encrypt all its communications when connecting to others. In some cases, a malicious user could modify the mobile agent code while it is running on its device intentionally; this may lead to unpredictable results and should be prevented. Thus, it is important to ensure the code integrity of the mobile agent and provide a mechanism to validate the agent's code integrity before starting its execution [30]. Furthermore, it is necessary to provide an authentication mechanism to trust an incoming agent in a wireless environment; this mechanism limits the access of mobile agent to mobile resources (memory, CPU, file system) to avoid misuse of these resources.

Some of the architectural issues in the mobile agent platform include location discovery services: the node's name discovery services and tracking name service [30]. The discovery service provides agents with appropriate mechanisms to find nodes' locations. This service is important because the mobile agent works in an open environment where devices can appear or disappear at any time. The node's name discovery service provides the ability to discover reachable nodes and whatever are their available services. The importance of this service is to allow an agent to detect services of other agents. The tracking of mobile agents and name services provides the ability to track the node locations efficiently, which allows communication with other agents.

There are many advantages of using agents in m-learning process for creating distributed systems. The mobile agent has the ability to move in different environments which is an important characteristic due to different mobile devices environments, capabilities and limitations. In addition, a mobile agent works independently and thus can control its own actions. Moreover, a mobile agent has independent knowledge and management; it can solve a given problem and can be adapted to the surroundings on its own. Additionally, it can operate at user's request under independent conditions and make plans for its own actions. Another benefit of mobile agents is a study-ability, which refers to the ability of the agent to adjust its own actions by the information it gathers from its surrounding communication [29]. Moreover, mobile agents can reduce the network load by allowing users to package a conversation and dispatch it to a destination host where interactions take place locally. This gives a great benefit to an m-learning system due to wireless communication limitations [31]. Because of these characteristics, building system architecture design for m-learning based on a mobile agent would be a good solution.

### 2.4. The Collaborative Mobile-Learning Prototype Architecture Model

Black and Hawkes [32] proposed a model for a collaborative mobile learning. The goal of this model is to create a prototype interface for mobile interaction among users using a Personal Digital Assistant (PDA). The model target domain is reading–comprehension and question–answer to provide collaborative interaction interface for mobile learning tasks by enabling textual or voice input. The model is a client-server architecture. The server-side components are Image Manager Servlet, File Manager Servlet, Message Manager Servlet and Session Manager Servlet. The Image Manager Servlet is responsible for receiving requests for image files from the client, loading those files, and sending files back to the client. The File Manager Servlet is responsible for file management tasks. The Message Manager Servlet is responsible for coordinating the sending of messages between clients. The Session Manager Servlet is responsible for initiating and management of the users' session.

The client-side components are the Interface Manager, MIDlet Session Manager, and local cache. The Interface Manager is the application interface managing the screen real-estate, text size; ease of movement, MIDlet Session Manager keeps track of changes in user data as the user moves through the session as well as the current state of some features of the interface. The local cache is used for storage of data that is used during the session. The system process flows begin by a user entering his/her ID for authentication purpose, and then the system presents the user with the lesson screens (i.e., story pages to be read). After reading is completed, the questions arise to be answered. The questions are checked for correctness, and if there are any incorrect answers, the user is allowed to collaborate with his/her partners, and in this case, icons of chat and group workbooks that are enabled.

### 2.5. Mobile Computer Support Collaborative Learning Models

Zurita and Nussbaum [11] introduced a model that supports collaborative mobile learning. The model is based on mobile computing through a wireless network using PDAs for children in the second year of elementary School. This model divides the Learners into groups that can move within their environment to make social interactions. Collaborator groups communicate among themselves and with other groups using the wireless communication network. The mobile can intercommunicate bi-directionally and wirelessly among them ensuring movement property for CSCL. Figure 2 illustrates how users can physically collaborate (face-to-face) in the dark gray area, where members of a group can wirelessly communicate in the light gray area.

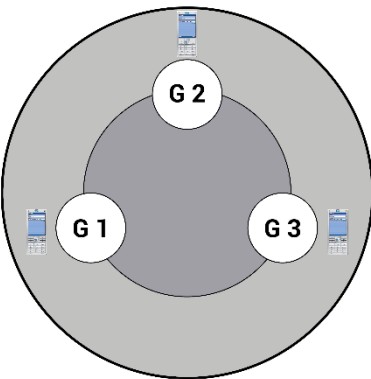

**Figure 2.** Computer Supported Collaborative Learning (CSCL) Mobile Model.

In addition, it is possible to form different groups when there are many students in a classroom. A node can join with one subgroup and exchange information to support the dynamic group reconfiguration task. The participant does not need to exchange data with all subgroup members, but only of one single group. Figure 3 shows a model in which the gray square represents the CSCL environment supported by mobile computing.

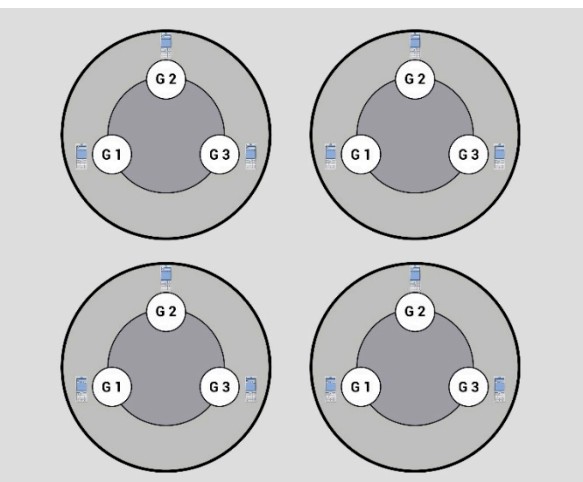

**Figure 3.** CSCL mobile system.

Reychav and Wu [21] proposed a research model that sheds light on collaborative mobile learning by examining the relationships between the learning process and the learning satisfaction, perceived understanding and performance, especially the role of individual learning in groups. Different experiments were conducted for this research model to understand the collaborative mobile learning process and the learning impact with mobile tablets. The study revealed that the performance

and satisfaction with texts is higher with mobile groups, while videos are more influential for individual learning.

The above approaches present the mobile collaborative m-learning architecture based on the client-server. The next section discusses the proposed model design for an interactive m-learning environment based on mobile agents as a basic step for developing a complete collaborative m-learning architecture.

## 3. M-Learning Model Design Based on Agents

The interaction between the learner and the teacher in the m-learning environment is very different from the real classroom. In a classroom setting, the learner can use eye expressions, sounds, body gestures, or express his/her opinions directly and indirectly. However, in the m-learning environment, the interactions between the learner and teacher are made using the mobile screen, communicating over networks, and are not limited by space and time. This section introduces a design model of interactive m-learning environment between learners and teachers as a basic step for developing an efficient collaborative learning model.

The proposed model consists of two main components: the learner agent and the teacher agent as shown in Figure 4. The learner agent has two elements; a learner profile and a mobile course. The teacher agent communicates with the learner agent to keep track of the student's learning status. The following subsections present the design of these agents.

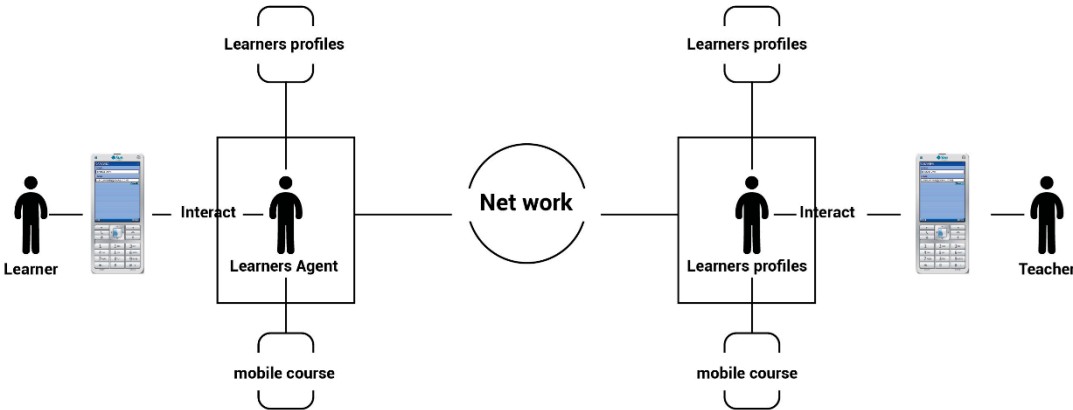

**Figure 4.** M-learning environment based on agents.

### 3.1. Learner Agent Design

A learner agent is deployed on the learner's mobile device in the m-learning environment. It maintains the learner personal information and monitors the learner progress for the respective learner and as shown in Figure 4. The learner interacts with the m-learning environment using a mobile user interface, and the learner agent observes the learner activity and updates the learner profile (achievement, understanding, and performance) to enhance the learning development process.

### 3.2. Teacher Agent Design

The teacher agent helps teachers to guide the learners study. It exchanges information with the learner agent in order to give the teacher an evaluation report about the learner's progress, and then the teacher evaluates the status of the learner and give a feedback to the learner by informing the teacher agent to notify the learner agent about teacher instructions. The teacher agent is deployed on the teacher mobile device. It analyzes the learner's information to give informative feedback. In addition, the teacher agent decides which learners understand the learning tasks and it gives clarifications. It can guide the learner for learning difficulties of his/her study. In addition, it can choose a course from mobile courses for discussion to enhance understanding. If there is any learner that does not follow

the teacher agent's instructions, the teacher agent can ask for help from the learner agent. The teacher agent will send a request to the relative learner agent. The learner agent will forward the teacher instruction to the learner for informing him/her about the required study.

The following steps summarize the m-learning environment based on agents:

1.  The learner agent observes learner activity (i.e., learner solves some exercise).
2.  The learner agent informs the teacher agent about the learner's progress (i.e., which topic was understood or misunderstood).
3.  The teacher agent guides the learner (i.e., give the teacher instructions, direct students to select the needed course).
4.  The teacher agent contacts the teacher to inform him/her about the learner's progress report.
5.  The teacher gives instructions to the learner through teacher agent.
6.  The teacher agent informs the learner agent about new instructions.
7.  The learner agent uses new instructions to enhance the learner's progress.
8.  The teacher agent monitors the progress of the teacher's instructions

## 4. Collaborative Mobile-Learning Model Design Based on Agents

Section 3 presented the proposed model design for the interactive m-learning environment based on mobile agents. Section 4 proposes a complete model for a collaborative m-learning architecture. The proposed model is based on the FRAME model discussed in Section 2 (see Figure 1). The FRAME model divides the mobile environment into three main parts: device (D) which represents mobile technologies, learner (L) which represents human learning capacities, and social (S) which represents social interaction. Based on the FRAME model, a collaborative m-learning model that covers the mobile learning environment is proposed. The proposed model divides the mobile environment into different parts; each part is responsible for one of the main aspects of the mobile environment in the FRAME model. The model components are device agent, learner agent, teacher agent, and social agent. These agents are interacting with each other to create a mobile learning environment as shown in Figure 5. For example, the social agent can interact with the learner agent to encourage the learner for more collaboration in the learning task. Similarly, the learner agent may interact with the device agent to find the location of another agent in the network. Furthermore, the teacher agent can interact with the learner agent to inform the learner of a new learning task or instructions. The design of the agents of the proposed model are described in detail in the following subsections.

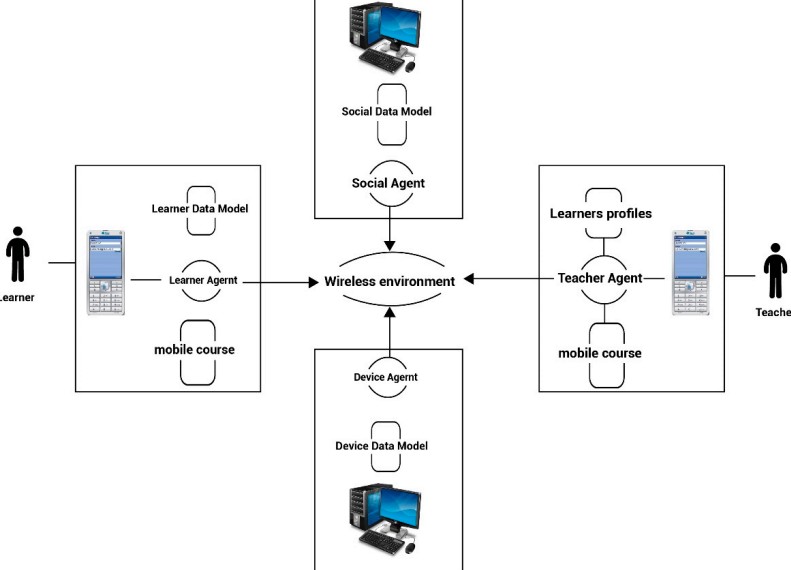

**Figure 5.** Collaborative mobile-learning model based on agents.

### 4.1. Learner Agent Design

The learner agent concerns with the learner aspects of the FRAME model. It is responsible for keeping track of the learner's profile that includes the learner development, learner evaluation, and learner characteristics. The learner agent in the collaboration model is connected with the learner data model in order to assess the learner's abilities and deficiencies, which will be constructed by the observation actions of the learner. The learner data model maintains the learner collaborative interaction that is used to evaluate the group collaboration. The learner data model is composed of the following parts: learner interactions, learner understanding, learner performance, and learner personality.

### 4.2. Teacher Agent Design

The teacher agent is concerned with the evaluation and it is guiding the learning process. It helps the teachers to follow the learners' progress and evaluation as described in Section 3. The learner agents inform the teacher agent about the learners' performance, and then the teacher agent will provide the teacher with the learner's evaluation reports. This helps the teacher to provide feedback to a learner and monitor their progress. The teacher agent can guide the learner during the learning process, give explanation, and provide exercises and discussions to enhance a learner's understanding. It gives a learner the instructions to be more collaborative and engage him/her to join to learning process.

### 4.3. Device Agent Design

The device agent is concerned with the device aspects of the FRAME model. It is responsible for all functions that are related to the mobile technologies, such as mobile platforms and networks. The device agent connects with the device data model that contains the device specifications and properties in order to configure the mobile learning environment and identify the different types of device terminals. The device data model is composed of the display formats, platform, operating system, screen size, network protocol, network bandwidth, and so on. Additionally, the device agent is responsible for providing the services that are required in the collaborative learning processes. One of these services is the discovery of mobile nodes as it helps the mobile device to determine the location of other nodes in the group, and how to reach these nodes. Another important service is the mobile discovery service (described in Section 2), which allows the mobile device to explore other services on other mobile devices. In addition, the device agent provides security services to the mobile device. For example, the mobile device needs to authenticate all messages exchanged with other mobile devices in the group. Therefore, the device agent may offer an encryption key for the group. The device agent is located at the server side, and other mobile agents can communicate with it (see Figure 5).

### 4.4. Social Agent Design

The social agent is concerned with the social aspects of the FRAME model. It is responsible for the social interaction and cooperation that includes managing group interactions, monitoring collaborative learning activities, evaluating group performance, and increasing collaborative learning within the group. The social agent connects with the social data model that contains all information related to the groups in order to assist the progress of the learner's group. For example, the social data model has the social profiles of learners and their classifications. The social data model can be stored in the database of the server, and the social agent can retrieve the collaborative data via queries. In the proposed model, the social agent is responsible for monitoring and assisting the learners in their collaborative activities. All learners are interacting socially using social agents. While the learner agent is responsible for monitoring and interacting with one learner, the social agent interacts with all learner agents of the group. For example, the social agent can interact with the learner agent to encourage the learner to collaborate more in group tasks. In addition, the social agent gives the learner agent the instructions to help the learner's collaborating with other learners. The social agent is located at the server side, and other mobile agents can communicate with it (see Figure 5).

*4.5. Analysis of Collaborative Mobile-Learning Model Based on Agent*

The following steps represent the interactive scenario between the components of the proposed model:

1.  The teacher interacts with the teacher agent to provide learners with the learning task. The teacher informs the teacher agent about new learning tasks.
2.  The teacher agent looks for the location (IP address) of the learner agent. To do this, it requests the needed information from the device agent, where the information is maintained in the device data model. The device agent checks its data model to retrieve the requested information, and then it provides the teacher agent with the learner agent location.
3.  The teacher agent sends the received teacher's learning task to the learner agent.
4.  When the learner agent receives the teacher's task, it informs the learner with the received teacher's task using the mobile user interface.
5.  The main goal of learner agent is to manage the learner's task with other learners. The learner agent gathers the information related to the learner by observing the learner's interaction, and then updates the learner data model.
6.  The responsibility of the social agent is to manage and monitor the collaborative learning between learners. Thus, the social agent collects the learner's interactions during the learning process. To accomplish this process, the social agent requests some information about the learner's agent location from the device agent. After it receives the node's location, it requests the learner data model from the learner agent (i.e., the social agent can move into the mobile device to collect the social information).
7.  To maintain the social information, the social agent updates the social data model.
8.  The social agent provides the teacher with an evaluation report showing the collaboration among learners. It sends the social information to the teacher agent, and then the teacher agent informs the teacher with the social information using the mobile user interface.
9.  Using the information gathered from the learner agent, the social agent provides the learner agent with the instructions that engage the learner in collaboration with other learners.

The main concern in this work is to create a collaborative learning model. In the next section, the role of the social agent in managing the collaboration among the learners is described with a focus on the social agent's monitoring task, and how the learner's collaboration can be analyzed to evaluate the group collaboration. Furthermore, the social agent's services that are required to maintain the social information are illustrated.

## 5. Collaborative Mobile Learning with Social Agent

The collaborative learning interactions between learners involve defining the problem and relevant parameters, suggesting solutions among learners, evaluating and elaborating the suggested solutions, choosing the best solutions, and finding the best decision. The collaboration interactions increase the social support, the positive school attitude, the positive attitude towards education, the on-task behavior, and the level of interpretation for the learners' group [33]. As described in Section 4, the social agent is responsible for maintaining social interactions among the learners to increase the collaborative learning within the group. This section demonstrates some functions that are responsible to the social agent in the learning process. In the proposed model, a collaborative learning environment is created to enhance the group interactions activity and increase the learner's social contribution in discussion, counter-suggestion, evaluation, and elaboration.

Section 5.1 demonstrates the social agent tasks that are important to create a collaborative learning environment. Section 5.2 describes the social services that are needed to accomplish the social agent task. Section 5.3 illustrates how data are organized in the social data model. This section also discusses

the social data interpretation and presents the social agent workflow. A collaborative learning scenario is demonstrated in Section 5.4.

*5.1. Social Agent Tasks*

To create a collaborative learning environment, the social agent is responsible to the following tasks:

1.  To monitor and assess the learner's collaborative activities.
2.  To evaluate the group collaboration performance.
3.  To provide a feedback to the teacher about the learner status and the group progress.
4.  To arrange the discussion between the learner groups.
5.  To provide feedback to the learners about their progress and contribution.
6.  To encourage the learner to undertake a learning task.
7.  To give the learner a tactic to enhance the collaboration with other learners.

The aforementioned tasks enable collaborative m-learning interaction among learners.

*5.2. Social Agent Services*

The goal of the social agent is to help both learners and teachers to collaborate in a learning environment. This goal requires the social agent to fulfil some tasks as described in Section 5.1. However, the social agent tasks need some services to be accomplished. This section describes these services, namely, assessment of quality group collaboration, assessment of group performance, and learner group participation.

5.2.1. Assessment of Quality Group Collaboration Service

The social agent is concerned with the interaction among learners; it analyzes the interactions among learners and enhances the collaborative learning activity. One of the tools used by the social agent to analyze the interactions among learners is the "Assessment Quality of Group Collaboration (QCA) Service." This service is important for the social agent to monitor and assess the learners, and encourage them to do the learning tasks. This service also analyzes the learners' interaction and provides feedbacks about learners' contribution in the learning task. The social agent uses QCA to assess the quality of the group collaboration service. It counts the frequency of certain types of content to be used in interpretations of constructs such as the learner behavior [34]. This procedure will be used to imply information about the learner's social skills according to the number of learner's participating in the learning task by assigning each contribution a specified code or category.

The QCA can be used to make an inference about unobservable constructs based on observable behavior by collecting a sample of behaviors from a specified domain. Each sample is assigned into a behavior category and the frequency of each used category reflects the differences in the implication. Additionally, by using the QCA, the social agent can interpret the quality of the group collaboration by collecting the learner interactions, and then assigning each interaction into some category, and counting the frequency of each category to make an interpretation about the group collaborations quality. Therefore, the categories for the learning problem need to be defined, and each interaction is classified into a category. Finally, an interpretation of the collected social data needs to be developed to make an evaluation for the quality of collaboration. The following steps describe the QCA Service:

1.  The learner agent observes learner interactions, then it updates learner data model.
2.  The social agent collects the learner's interactions (i.e., the social agent moves into a mobile device to collect the social information).
3.  The social agent classifies each interaction to one of defined classes (described in Section 5.3).
4.  The social agent makes interpretations and analysis on the social data using the QCA to assess the quality of group collaboration (described in Section 5.4).

5.      The social agent informs the teacher agent about the quality of the group collaboration.

6.      The teacher agent notifies the teacher about the quality of group collaboration.

### 5.2.2. Assessment of Group Performance Service

The social agent can measure the group performance by assessing the interpretation of learners' interactions. The social agent classifies all interactions received from the learner agent and assigns a rank for everyone (the rank can be a number or an interpretation). The social agent then assesses the rank of interpretations to measure the group performance. The interpretations of interactions rank can be assigned according to how close are the interactions to the learning goal. The learning goal can be the solutions to a problem, the overall teamwork on a project, a number of tasks that are done etc. In the proposed model, the group performance can be measured by assigning each class a rank. Then the frequency of each class is calculated and multiplied by its rank; then, all total ranks are added up to obtain the overall group performance. The social agent evaluates the group performance and informs the teacher about the evaluation to help the teacher to give a feedback for group members as shown in Figure 6.

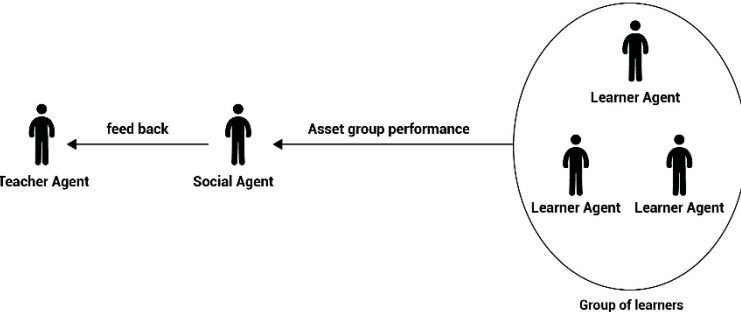

**Figure 6.** Assessment of group performance.

The following steps describes the Assessment of Group Performance Service:

1.      The learner agent observes the learner's interactions, and then it updates the learner's data model.

2.      The social agent collects the learner's interactions.

3.      The social agent classifies each interaction to one of the defined categories (described in Section 5.3).

4.      The social agent makes interpretations and analysis on the social data using QCA to assess the group performance (described in Section 5.4).

5.      The social agent informs the teacher agent about the group's performance.

6.      The teacher agent notifies the teacher about the group's performance.

### 5.2.3. Learner Group Participation Service

The learner group participation estimates the learner participations in the group interactions and shows how the learner contributes to the group task relative to all group members. This measurement is useful for the social agent to decide which learner should be encouraged on the learning task, and to give the learner a tactic to enhance his/her collaboration with other group members. The social agent can calculate the learner group participation by dividing the total number of his/her interactions on the total number of interactions of the whole group. It should be noted that the learner may not interact with other learners; he/she may only observe the activity of other learners. This mode of observation may be considered as a type of participation depending on the aims of a study. The following sequence steps describes the learner group participation service:

1.      The learner agent observes learners' interactions, and then it updates the learner data model.

2.      The social agent collects learners' interactions.

3. The social agent classifies each interaction to one of the defined categories (described in Section 5.3).
4. The social agent makes interpretations and analysis on the social data using the QCA to evaluate the learner group participation (described in Section 5.4).
5. The social agent encourages learners who did not contribute in the learning task by communicating with the learner agent.

### 5.3. Social Data Model

The social data model contains the social profiles of the learners and the interaction classifications. It maintains the classes of interactions to identify a class for each interaction as shown in Table 1. The goal here is to ensure that the classes of the interactions are combined with the behaviors that represent the learner interaction of that class (i.e., the class is a problem analysis). This is particularly important in the work of the QCA because this technique is essentially observational. In addition, the social data model maintains the learners' social profiles. The learner profile consists of the learners' interaction classes that resulted from the collaborative learning. The social data model stores this information into the learners' interactions sequence as shown in Table 2. The interpretation of the interactions sequence is discussed as follows.

**Table 1.** Classes of interactions.

| Class Type | Interactions Type (Construct) |
|---|---|
| Class 1 | Interaction 1, Interaction 2, interaction 3 |
| Class 2 | Interaction 4, Interaction 5, interaction 6 |
| Class 3 | Interaction 7, Interaction 8, interaction 9 |

**Table 2.** Learners' interactions sequence.

| Learner Number | Interactions Profile |
|---|---|
| Learner (1) | Class 1 – Class 1 - Class 3 – Class 4 – Class 2 . . . etc. |
| Learner (2) | Class 1 – Class 1 - Class 3 – Class 4 – Class 2 . . . etc. |
| Learner (3) | Class 1 – Class 1 - Class 3 – Class 4 – Class 2 . . . etc. |

The interpretation of the social data needs to define each interaction and assign it to a specific class. After doing this classifications, the social agent builds the learners interactions sequence (i.e., Class 1 – Class 1 - Class 3 – Class 4 – Class 2 . . . etc.) according to the learner's social interactions shown in Table 2. Then the social agent analyzes the learner's interactions sequence using the QCA to assess the quality of group collaboration, group performance, and learner group participation. To assess the quality of the group collaboration, the social agent constructs the learner's participations values as shown in Table 3. This table shows the learner class's participations number (N1, N2, and N3) and the total learners class participations number (Total N). The value N reflects the quality of group collaboration in the learning class (a class can be any skill such as problem analysis or problem evaluation).

**Table 3.** Learners' participations table.

| Learner /Class | Class 1 | Class 2 | Class 3 |
|---|---|---|---|
| Learner (1) | N1 | N2 | N3 |
| Learner (2) | N1 | N2 | N3 |
| Learner (3) | N1 | N2 | N3 |
| TOTAL | Total N1 | Total N2 | Total N3 |

To assess the group performance, the social agent constructs an interactions ranking as shown in Table 4. This table shows a class rank and the total learners' class participation number. The total rank

for a specific class reflects the group performance of this class, while the overall total rank reflects the overall group performance in all classes during the learning process.

**Table 4.** Learners' interactions ranking.

| Class | Rank | Interactions Number | Total Rank |
|-------|------|---------------------|------------|
| Class 1 | R1 | Total N1 | R1*N1 |
| Class 2 | R2 | Total N2 | R2*N2 |
| Class 3 | R3 | Total N3 | R3*N3 |
| **Total** | - | - | Overall total rank |

To assess the learner's group participations, the social agent calculates the participation ratio of the learner by dividing the learner class participations number (N1, N2, N3) over the total learners class participations number (Total N1, Total N2, Total N3) as shown in Table 5. These statistics can be utilized to analyze the learner's participations during the problem solving of the collaborative environment. For example, if R1 is low, then it can be concluded that the learner has low collaboration in Class 1. In this case, the social agent encourages the learner to make more contributions with other group members.

**Table 5.** Learners' participation ratio.

| Learner /Class | Class 1 N1/Total N1 | Class 2 N2/Total N2 | Class 3 N3/Total N3 |
|----------------|---------------------|---------------------|---------------------|
| Learner (1) | R1 | R2 | R3 |
| Learner (2) | R1 | R2 | R3 |
| Learner (3) | R1 | R2 | R3 |

Figure 7 illustrates the social agent's workflow. The social agent uses the class definitions in Table 1 to classify the learner's contribution and obtain the learner's interactions sequence, and then it counts the number of each class to make an interpretation. The interpretation is based on the comparison of the learner participation with the whole group participations. The following steps summarizes the social agent's workflow:

1. Determine a category for each interaction in the social data model.
2. Define each construct (interactions among learners) and assign it to a category.
3. Define interpretations for the social data using the QCA (described in Section 5.4).
4. The social agent collects the interactions among learners (i.e., discussion messages) by interacting with the learner agents. For this purpose, the learner agent maintains the learner data model to retrieve the learner interactions.
5. The social agent classifies each interaction to one of the defined categories.
6. The social agent makes interpretations and analysis of the social data using the QCA. It counts the frequency of classes in the learner interaction sequence to infer conclusions about the learner behavior.
7. According to interpretations resulting from the previous step, the social agent can monitor and assess the learners in the collaborative activities and provides a feedback to the teacher.

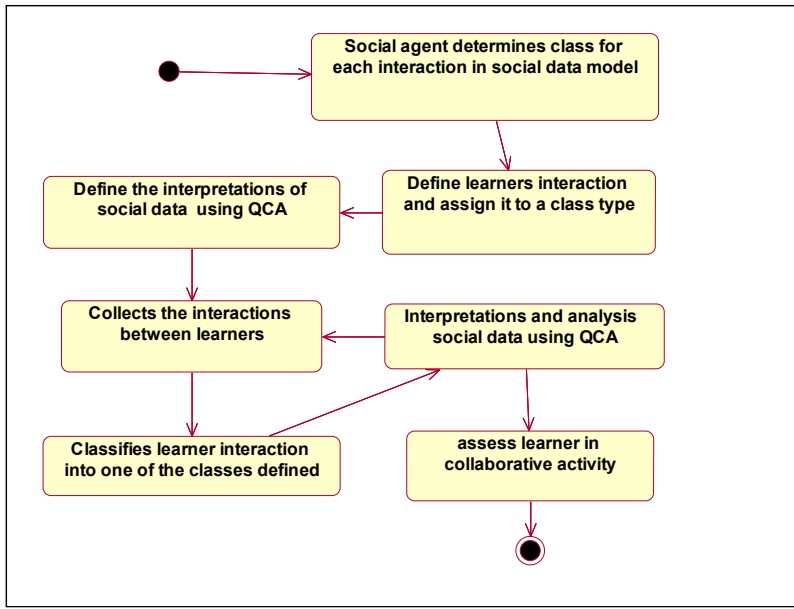

**Figure 7.** The social agent workflow.

*5.4. Collaborative Learning Scenario*

In this section, a collaborative learning scenario is described to demonstrate how the social agent is used in the problem solving of collaborative m-learning. The scenario consists of a group of learners and a teacher assigning a task to the group. The group goal is to collaborate to solve the given task. The interactions of the learners' group are defined as discussions among the learners. The social agent evaluates the group discussion to assess the quality of the group collaboration, group performance, and learner group participation, and then it returns feedback to the teacher and group members.

5.4.1. Development of Assessment Quality of Group Collaboration Service

The QCA service is used to understand the discussions among the learners on the syntax of words in a sentence by analyzing the interactions among the learners. The first step is to define the class of interactions (Table 1) in the social data model. Therefore, the interaction classes need to be identified with a respective construct (interaction).

Poole and Holmes [35] developed a Functional Category System (FCS), which is an interaction classification scheme to study the collaborative interactions in problem-solving contexts; they provide a classification for each interaction unit in seven categories:

1.  Problem Analysis: defined as statements that state the causes behind a problem (e.g., "I think in this project we are supposed to create a model . . . ").
2.  Problem Critique: defined as statements that evaluate the problem analysis statements (e.g., "what we need to do is to create this model . . . ").
3.  Orientation: defined as statements that attempt to orient or guide the group in the learning process, and to reflect an evaluation of a group's process or progress (e.g., "my opinion is to draw on the notes about everything").
4.  Criteria Development: defined as statements that focus on criteria for decision-making or on general parameters used in solutions (e.g., "we need to get a basic model with its components . . . ").
5.  Solution Development: defined as suggestions of alternatives, ideas, proposals for solving the problem, and statements that provide details or elaboration on a previously stated alternative. It provides ideas or further information about alternative solutions (e.g., "I suggest to create a simulation for this model").

6.   Solution Evaluation: defined as statements that evaluate alternatives and give reasons for the evaluations, including statements that agreed or disagreed with the criteria development or solution suggestion statements. The statements in this category can be a decision in its final form or ask for a final group confirmation of the decision (e.g., "I agree with your solution").

7.   Non-Task: defined as statements that do not have anything to do with the decision task (e.g., "let us take a break!").

The classes of interactions to represent the problem solving are built based on FCS. Table 6 contains the Functional Category System (FCS) classes and interactions (keywords) that can be used in the classification. The keywords are defined based on the problem's possible interactions among learners during the learning task. The full FCS class interactions used are shown in Table A1.

**Table 6.** Functional Category System (FCS) classes of interactions.

| Class Type | Interactions (Keywords ) Observed |
|---|---|
| Problem Analysis (PA) | I think, analysis |
| Problem Critique (PC) | How, what, when |
| Orientation (OO) | Find, go to, opinion |
| Criteria Development (CD) | parameters, variable |
| Solution Development (SD) | Solution, suggest, idea, propose, task |
| Solution Evaluation (SE) | Evaluate, I agree, disagree |
| Non-Task (NT) | Break, other non-learning statements |

The learner interactions are classified according to the classes of interactions (e.g., Table 6) defined in the social data model. The social agent builds a FCS Learners interactions sequence based on learner interactions (Table 7). For example, Learner (1) has the interactions sequence (PA–PA–OO–CD–PA–SE–NT–SE–SD). The analysis of this sequence determines the quality of the learner collaboration in the group.

**Table 7.** FCS learners' interactions sequence.

| Learner Number | Interactions Sequence |
|---|---|
| Learner (1) | PA–PA–OO–CD–PA–SE–NT–SE–SD |
| Learner (2) | PA–PC–CD–PA–PA–SE–SD–SE–SD |
| Learner (3) | NT–PC–PC–CD–NT–SE |

The social agent classifies each interaction among one of the defined FCS classes. Then it makes interpretations of the social data using the QCA to count the frequency of learner participation classes and construct FCS learners' participation as shown in Table 8. The FCS learners' participation contain the learner participations class number and the total learners' participation class number for each class. These statistics reflect the group quality of collaboration (i.e., total PA = 6). These values are used to evaluate the group collaboration and compare the evaluated collaboration with other groups in order to enhance the group collaboration in a proper way.

**Table 8.** Sample of FCS learners' participation.

| Learner/Class | PA | SE | CD |
|---|---|---|---|
| Learner (1) | 3 | 2 | 0 |
| Learner (2) | 3 | 2 | 1 |
| Learner (3) | 0 | 1 | 1 |
| TOTAL | 6 | 5 | 2 |

### 5.4.2. Development of Assessment Group Performance Service

To assess the group's performance, the social agent constructs an FCS learners' interactions ranking as shown in Table 9. The overall total rank (the value 25 in Table 9) reflects the overall group's performance, while the total rank for any specific class reflects the group class performance in the collaborative learning process. As illustrated in Table 9, the class SE has the rank value three and the total class participation number is five, and so, the total rank of the SE class is 15 (3 × 5). Comparing the SE class total rank with other classes, the SE class has the best total rank and the CD class have the worst total rank (Total Rank = 4). From this observation, it can be concluded that the group performance of SE is better than CD. In this case, the teacher will focus to grow the CD skills among the learners. The total rank reflects the collaborative group performance in the m-learning. If a social agent notices this total is low compared with other class types, then it notifies the group members to obtain more contributions for that class or skill.

**Table 9.** Sample of FCS learners' interactions ranking.

| Class | Rank | Number of Interactions | Total Rank |
|-------|------|------------------------|------------|
| PA    | 1    | 6                      | 6          |
| SE    | 3    | 5                      | 15         |
| CD    | 2    | 2                      | 4          |
| Total | -    | -                      | 25         |

### 5.4.3. Development of Learner Group Participation Service

The social agent calculates the participation ratio of the learner classes by dividing the learner participation class number over the total learners' class participation number as shown in Table 10. According to the ratios of the learner participations in Table 10, the social agent encourages, for example, Learner (1) to make more collaboration in the criteria development (CD = 0%). Similarly, the social agent encourages Learner (3) to collaborate more in the problem analysis (PA = 0%). Moreover, the social agent informs the teacher agent about the learner group participation. The next section discusses the implementation and simulation of the proposed architectural model using the Java Agent Development Framework–Lightweight Extensible Agent (JADE-LEAP) Platform.

**Table 10.** Sample of FCS learners' participation ratio.

| Learner/Class | PA N1/ Total N1 | SE N2/Total N2 | CD N3/Total N3 |
|---------------|-----------------|----------------|----------------|
| Learner (1)   | 50%             | 40%            | 0%             |
| Learner (2)   | 50%             | 40%            | 50%            |
| Learner (3)   | 0%              | 20%            | 50%            |

## 6. The Implementation of Collaborative Mobile-Learning Model Based on Agents

This section discusses the implementation and simulation of the proposed m-learning architectural model. The proposed model is implemented using Java 2 Micro Edition (J2ME) with JADE-LEAP services enabling agent interaction. The proposed architecture of the system is composed of multi-agents, namely, the device agent, the learner agent, the teacher agent, and the social agent. The authors deployed the agents on mobile devices and experimented with the platform in a real environment. The results obtained show that the collaborative m-learning should be used to supplement classroom lectures, enhance the leaner's skills in problem solving, increase the learners' knowledge, and encourage the learners to participate more in the learning tasks. In addition, the system is easy to use and access, well-organized, convenient, and facilitates the learning process. The following sub-sections present in details the implementation of the proposed m-learning architectural model.

### 6.1. Java Agent Development Framework (JADE) Platform Overview

Java Agent Development Framework (JADE) is a middleware that facilitates the development of multi-agent systems [36]. JADE provides simple and reliable access to services over any network, independent of platform, protocol, or application technology [37]. The JADE runtime environment is called a container. Each container can manage several agents. One container needs to be set as the main container and must always be active in the platform and all other containers register with it. Figure 8 shows two normal containers registered with the main container. Any container can manage more than one agent (e.g., Container 2).

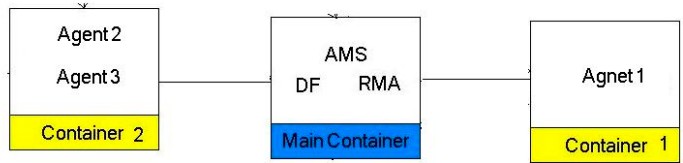

**Figure 8.** Java Agent Development Framework (JADE) container and platform.

The main container differs from other containers as it holds two special agents: The Agent Management System (AMS) and the Directory Facilitator (DF). The AMS is responsible for (1) ensuring that each agent in the platform has a unique name and (2) managing the authority in the platform. The DF is responsible for providing a Yellow Pages service by means of which an agent can find other agents providing the services required to achieve its goal. The Remote Monitoring Agent (RMA) provides a graphical console to monitor and control the platform, and it also manages the life-cycle of the agents [36].

### 6.2. JADE-Lightweight Extensible Agent (LEAP) Overview

JADE powered by a Lightweight Extensible Agent Platform (JADE-LEAP) is a platform that can be deployed on a wide range of devices varying from servers to Java-enabled cellphones. The JADE-LEAP platform can be executed on handheld devices supporting J2ME such as the great majority of Java-enabled cellphones. For this reason, JADE-LEAP is chosen to build the proposed architecture. Figure 9 shows the JADE-LEAP platform runs on Mobile Information Device Profile (MIDP).

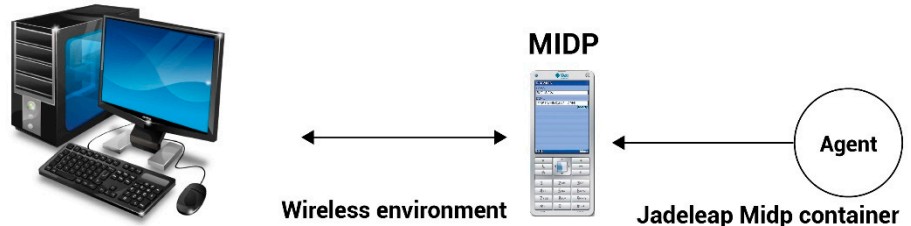

**Figure 9.** JADE-Lightweight Extensible Agent (LEAP) Platform Using Mobile Information Device Profile (MIDP).

### 6.3. Implementing the Proposed Collaborative Architecture Using JADE-LEAP

In this part, an agent-based system for the collaborative mobile learning using JADE-LEAP environment is constructed. The architecture of the proposed system is composed of a multi-agent system. Figure 10 shows the architectural design for the proposed collaborative mobile-learning model based on agents using JADE-LEAP platform. The architecture is composed of four agents: device agent, learner agent, teacher agent, and social agent. According to JADE-LEAP specifications, every agent must be located in a container. The DF and AMS agents work as device agents in our model. As illustrated in Figure 10, there are three learner agents located in three different containers, Container 1, Container 2, and Container 3. Each learner agent is registered with the main container using DF and

AMS agents. The DF and AMS agents keep track of the agent services and provide an Agent Identifier (AID) to enable the agent communication with other agents. Every learner agent receives the AIDs of other learner agents from the DF to interact with these learner agents. For example, the teacher agent registers with the main container to enable other agents to communicate with it as illustrated in Figure 10.

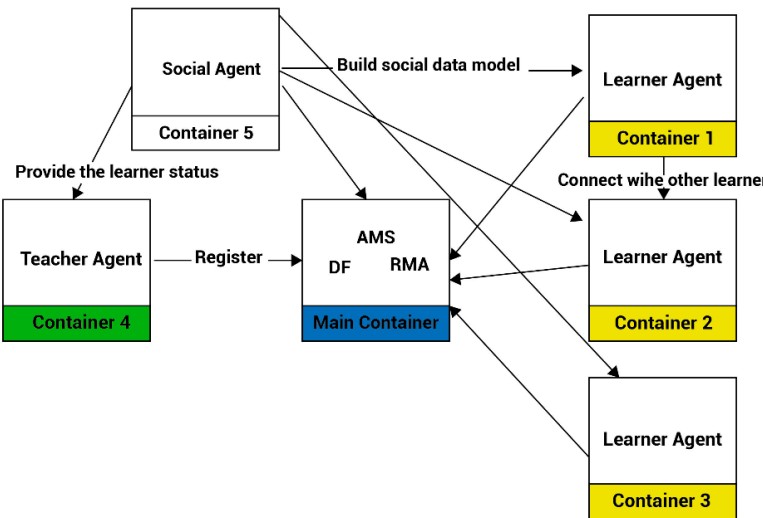

**Figure 10.** Proposed architectural design for collaborative mobile learning platform using JADE-LEAP.

The social agent interacts with other agents to keep track of all collaborative activities among the group members. It builds a social data model by interacting with all learner agents and downloading their AIDs. In addition, it communicates with the teacher agent to inform it about the learners' status. Figure 11 illustrates the architectural design for the collaborative mobile learning multi-platform based on agent using JADE-LEAP platform. In this multi-platform, the learner agent in Container 3, for example, can interact with the learner agent in Container 1 that is located in a different platform by communicating with its main container. This is possible because the two main containers are connected to each other.

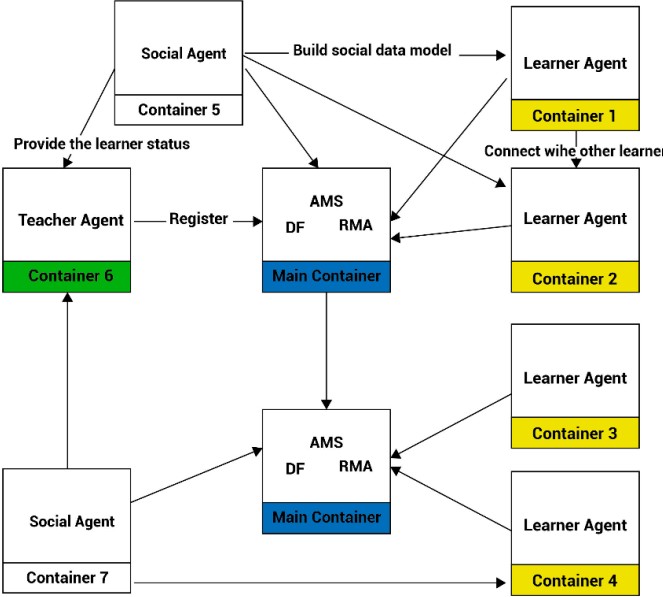

**Figure 11.** Proposed architectural design for collaborative mobile learning multi-platform using JADE-LEAP.

### 6.3.1. Implementation of Learner Agent

The learner agent (described in Section 4) is responsible for tracking the learner profile that includes the learner development, learner evaluation, and learner characteristics. Figure 12 shows a graphical user interface (GUI) for a mobile learner agent.

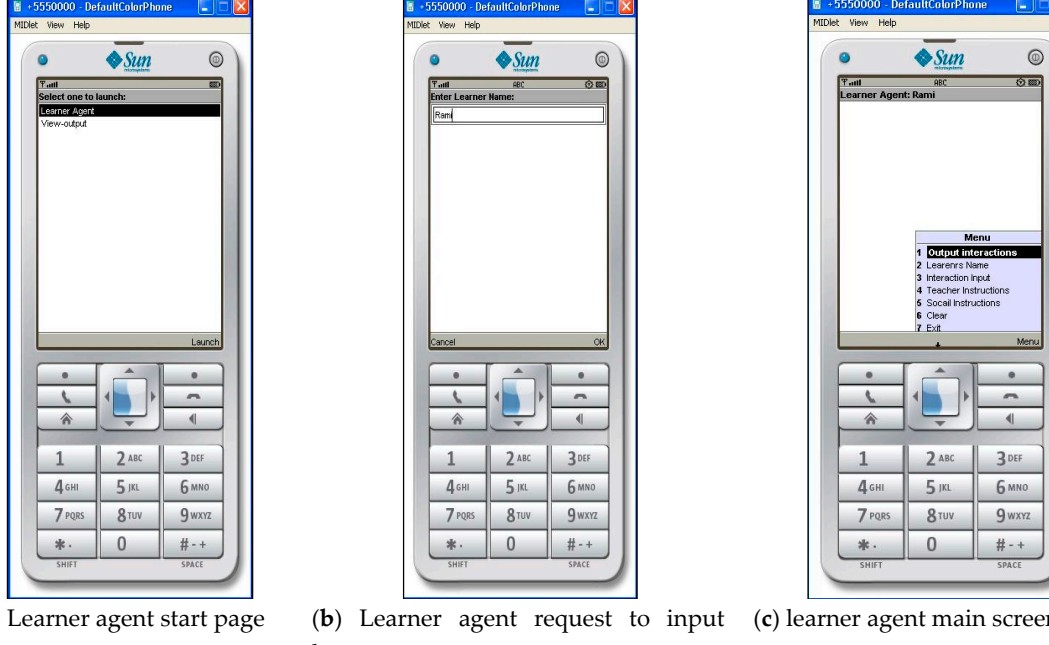

(**a**) Learner agent start page     (**b**) Learner agent request to input learner name     (**c**) learner agent main screen

**Figure 12.** Mobile Learning graphical user interface (GUI)—the learner agent.

Figure 12a shows the learner agent starting page that contains two selection items: learner agent and view-output. View-output contains the log of the agent and cannot be viewed during agent running due to limitation of mobile device screen and memory. The learner agent item will pass to the next screen that requests to input the learner name as shown in Figure 12b. The learner agent main screen is shown in Figure 12c. It contains the following selection items:

1. Output interactions (Figure 13a).
2. Learner name (Figure 13b).
3. Interaction input (Figure 13c).
4. Teacher instruction (Figure 14a).
5. Social agent instructions (Figure 14b): enables the learner to read the social agent instructions and notifications during the learning process.

The learner agent receives the instructions from the social agent to inform the learner about the learner status and what are the skills the learner should improve as illustrated in Figure 14.

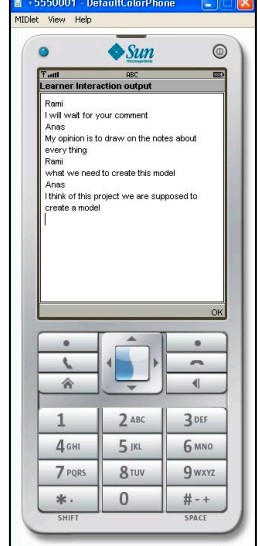
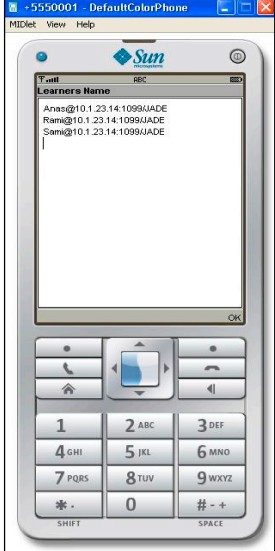
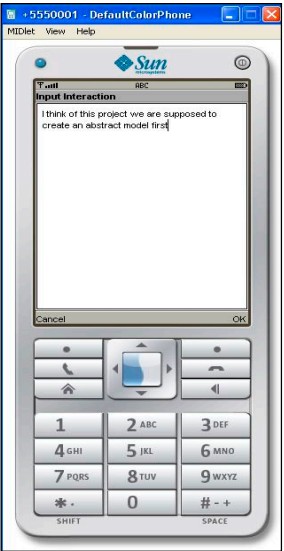

(**a**) Learner agent shows output interactions screen

(**b**) Learner agent shows learner name screen

(**c**) Learner agent interactions input screen

**Figure 13.** Mobile learning GUI—the learner agent shows learner output interactions, learner's name, and input interactions.

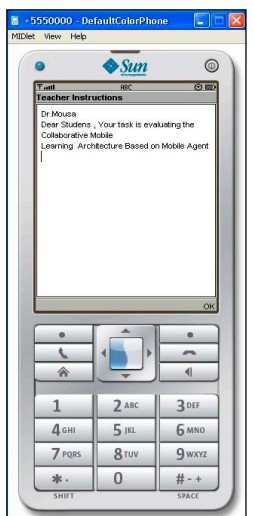
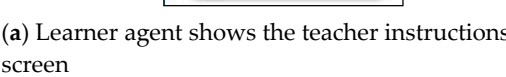
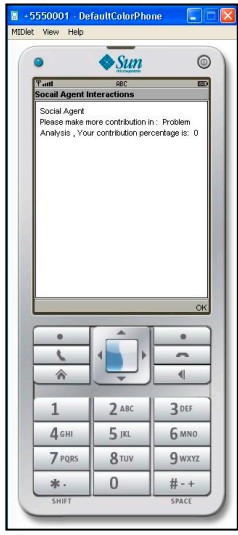

(**a**) Learner agent shows the teacher instructions screen

(**b**) Learner agent shows the Social Agent instructions screen

**Figure 14.** Mobile learning GUI—the learner agent shows social agent instructions and teacher instructions screens.

### 6.3.2. Implementation of Teacher Agent

The social agent informs the teacher agent about the learner's information to give the teacher an evaluation report about the learner's progress. The teacher evaluates the status of the learner and gives a feedback to the learner by informing the teacher agent to notify the learner agent about the teacher instructions. Figure 15 shows the GUI of the teacher agent. The teacher agent main screen (Figure 15c) contains the following items:

1. Learners' participation ratio (Figure 16a).
2. Learner name.
3. Instructions Input (Figure 16b).

4.  Learners Participations Table (Figure 16c): shows the learners' participation during the learning process as described in Section 5.
5.  Teacher instruction: enables the teacher to read the previous teacher instructions and notifications.

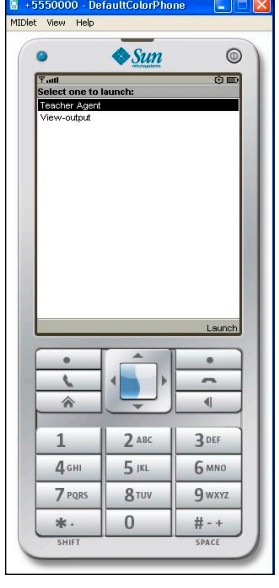
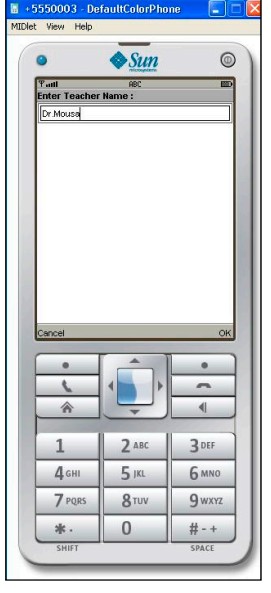
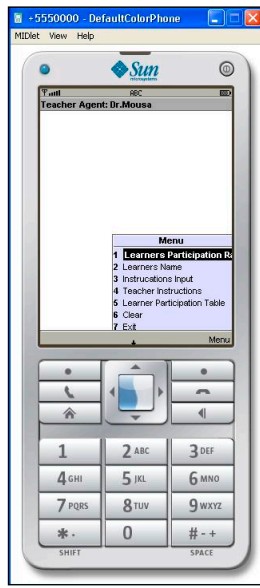

(**a**) Teacher agent start page      (**b**) Teacher agent request to input teacher name      (**c**) Teacher agent main screen

**Figure 15.** Mobile learning GUI—teacher agent on mobile device.

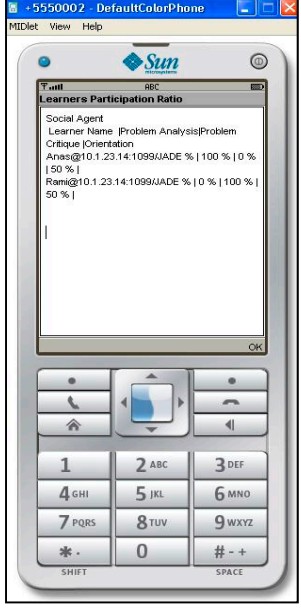
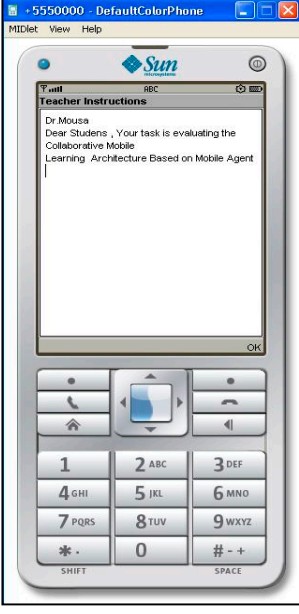
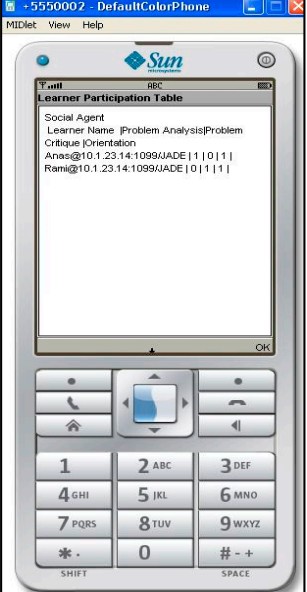

(**a**) Teacher agent shows Learner participation ratio      (**b**) Teacher agent input instruction screen      (**c**) Teacher agent shows Learner participation table

**Figure 16.** Mobile learning GUI—teacher agent on mobile device.

6.3.3. Implementation of Social Agent

The social agent is responsible for analyzing the social interactions among the learners to increase the collaborative learning among the group members. Figure 17 shows the GUI of the social agent.

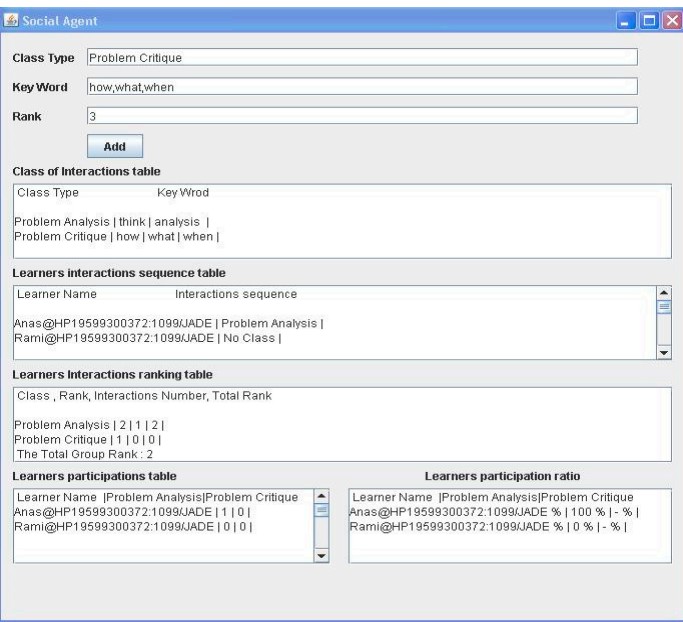

**Figure 17.** GUI of the social agent.

The social agent items are described as follows:

1. Input fields for class type, keyword, and rank. The teacher or module coordinator can define these values in the system.
2. Class interaction table as described in Section 5.
3. Learners' interaction sequence table: shows the learners' participation sequence table during the learning process as described in Section 5.
4. Learners' interaction ranking table: shows the learners' participation ranking table during the learning process as described in Section 5.
5. Learner's participation ratio as described in Section 5.
6. Learners participation table as described in Section 5.

The social agent builds the social data model by communicating with the learner agent and collecting the learner data model. The social data model consists of the classes interactions table, learner interactions sequence table, learner interactions ranking table, learner participations table, and learners' participation ratio. After collecting the social data, the social agent analyzes the information to build the learner participation table and learner participation ratio. Then the social agent communicates with the teacher agent to inform it about the learner's status.

6.3.4. Learner Agent and Teacher Agent Implementation Using JADE–E-Learning Extension

In this part, the authors implemented other versions of the learner and teacher agents using JADE (Figure 18). These versions can be used to extend the system to be deployed on devices that support J2SE, and so the system can be also used in the E-Learning environment.

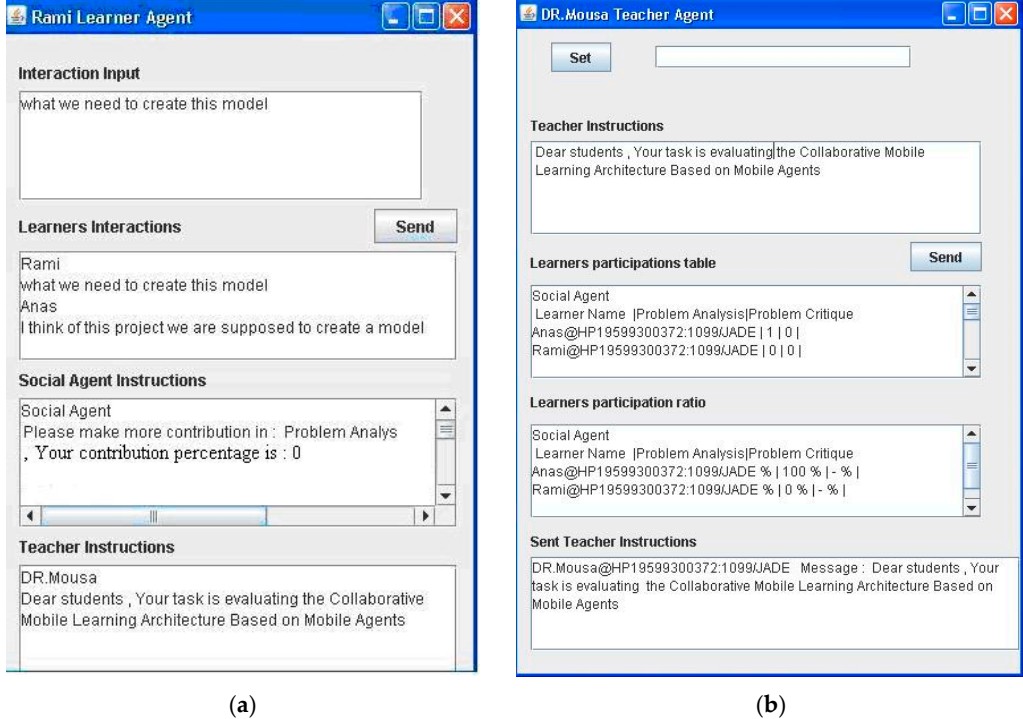

**Figure 18.** The GUIs of (**a**) the learner agent and (**b**) the teacher agent.

*6.4. Simulation of the Interactions between Agents*

This part presents sample interactions between the agents of the proposed system. Figure 19 illustrates a simulation to the proposed model using JADE Sniffer (Sniffer is used to flow the interactions among agents). As shown in Figure 19, the yellow arcs at lines 1 and 2 represent a learner agent (A) requesting an AID from the device agent in order to establish a communication. The grey arc at line 3 represents the case when the social agent informing the teacher agent about the learners' collaborations status. The grey arcs at line 4 and 5 represent the case when the social agent asks the learner agent for the learner data model to build the social data model as described in Section 5. The red and black arcs at lines 6 and 7 represent the case when the learner agents (A) and (B) respond to the social agent for its request. The other arcs (purple, blue, and green) represent the agents' AID updates.

Figure 20 illustrates the teacher agent interactions with other agents. The grey arcs at lines 1 and 2 represent the case when the teacher agents propose a task to the learner agents (A) and (B). The purple and blue arcs represent the requests and responses of the teacher agent for the AID learner and social agents.

Figure 21 shows the simulation of interactions between the learner agents. The grey arcs at lines 1 and 2 represent the case when the learner agent (B) interacts with other learners of the same group.

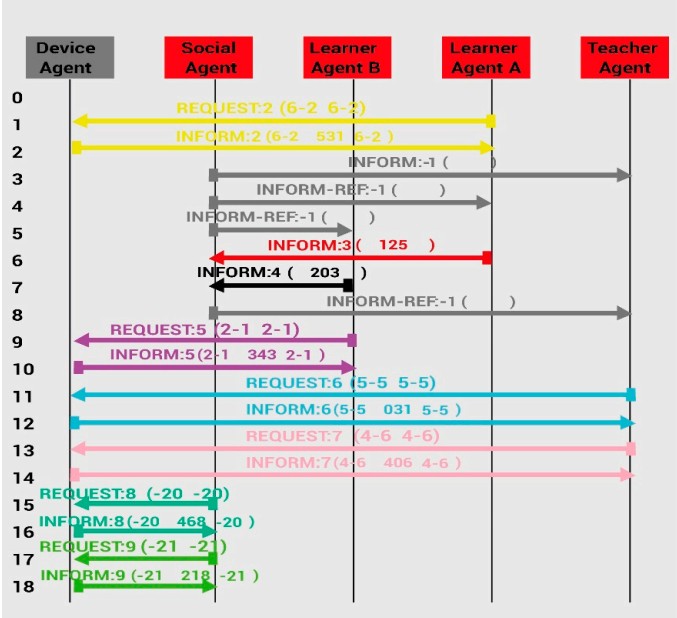

**Figure 19.** Simulation of interactions between agents.

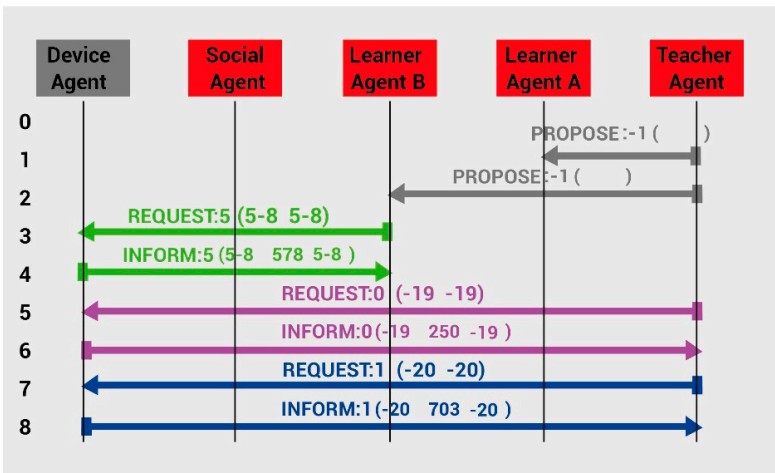

**Figure 20.** Simulation of interactions between teacher agents and other agents.

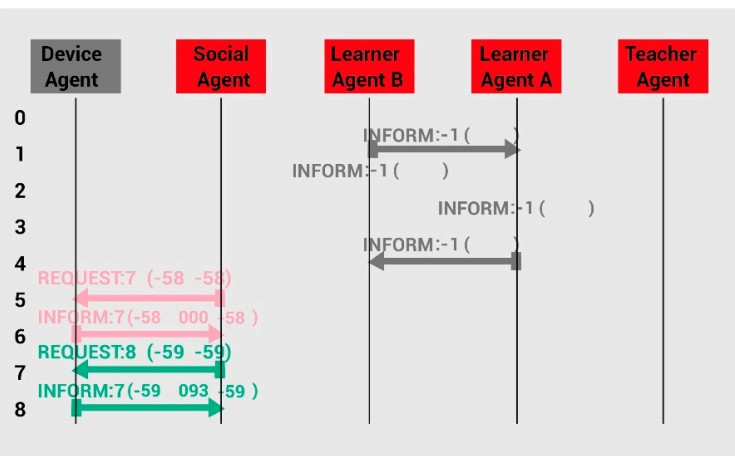

**Figure 21.** Simulation of interactions between learner agents.

## 7. A Case Study for Collaborative Mobile Learning Model

To evaluate the proposed model, it was tested by a group of students at the University of Jordan. The questions are selected from "Data Structure" course that is taken by all undergraduate IT students at the university. The selected students are registered in "Database" and "Artificial Intelligence" courses as these two courses have the "Data Structure" course as a pre-requisite. Section 7.1 presents the experiment methodology used in the test and the experimental results obtained from a distributed questionnaire, Section 7.2 illustrates the analysis of interactions between students during the test. Section 7.3 analyzes the experimental learner interactions.

### 7.1. Experiment Methodology and Results

To conduct the model experiments, two full sections of 26 students, of King Abdullah II School for Information Technology located at the University of Jordan, were invited to use the developed collaborative m-learning system. The standard section size ranges between 20 to 30 students, therefore this number reflects the most realistic scenario in such settings. The University of Jordan is the largest and oldest institution of higher education in Jordan located in Amman. The selected students were asked to complete a questionnaire after they use the m-learning model. The students were divided into four collaborative groups (Group 1, Group 2, Group 3, and Group 4) where 4 to 8 students in each group cooperated to solve the given problem. Students in each group were asked to collaborate to solve a question taken from the "Data Structure" course (refer to Appendix A for problem description and the used FCS class interactions table). The questionnaire measures the students' acceptance and perceptions of the effectiveness of the collaborative m-learning system. On a five-point scale (ranging from 1 = "strongly disagree" to 5 = "strongly agree"), students were asked to indicate their response to 21 statements about the collaborative m-learning system. Frequencies and descriptive statistics such as mean and standard deviation (SD) values for each question of the questionnaire were calculated. Table 11 shows the main findings of the questionnaire, which are the mean and SD values for each question of the questionnaire. This table has three columns; the first is the question number (Appendix B lists the questions), the column labeled Mean shows the mean value of each question, and finally, the third column labeled SD shows the standard deviation value of each question.

**Table 11.** Questionnaire findings. Scale: 5 = Strongly Agree; 4 = Agree; 3 = Neutral; 2 = Disagree; 1 = Strongly Disagree.

| Question Number | Mean | SD | Question Number | Mean | SD | Question Number | Mean | SD |
|---|---|---|---|---|---|---|---|---|
| Q1 | 3.88 | 1.01 | Q8 | 3.73 | 0.90 | Q15 | 3.69 | 0.61 |
| Q2 | 3.73 | 0.90 | Q9 | 2.69 | 0.72 | Q16 | 3.85 | 0.60 |
| Q3 | 3.73 | 0.76 | Q10 | 3.62 | 1.08 | Q17 | 3.65 | 0.73 |
| Q4 | 3.92 | 0.67 | Q11 | 3.50 | 1.08 | Q18 | 3.65 | 0.62 |
| Q5 | 2.58 | 0.97 | Q12 | 3.88 | 0.97 | Q19 | 3.85 | 0.72 |
| Q6 | 3.31 | 1.26 | Q13 | 3.96 | 0.85 | Q20 | 3.77 | 0.70 |
| Q7 | 3.15 | 1.46 | Q14 | 3.54 | 1.01 | Q21 | 3.62 | 0.84 |

Overall, the students perceived that the proposed collaborative m-learning application is easy to use (3.88, SD = 1.01), well organized (3.73, SD = 0.90), facilitates learning (3.73, SD = 0.73), should be used to supplement lectures (3.73, SD = 0.90), and convenient and easy to access (3.92, SD = 0.67). On the other hand, students have different thought about using the m-learning application to replace the traditional lectures (3.15, SD = 1.46) and how the m-learning matches the human contact that a teacher provides (3.31, SD = 1.26). However, students felt the proposed model enhances the problem-solving skills (3.96, SD = 0.85). In addition, students expressed that they like the combination of both lectures and m-learning (3.88, SD = 0.97), and they think the collaboration among students in the m-learning is useful (3.85, SD = 0.72). Additionally, students agree that collaborative m-learning increases their

knowledge rather than learning individually (3.62, SD = 0.84) and the social agent encouraged them for more participation in learning (3.77, SD = 0.70).

*7.2. Analyzing the Experiment Learners' Interactions*

This section shows the analysis of the social interaction between the learners. Table 12 presents the FCS interactions sequence for the first collaborative group (Group 1). It can be noticed that the learners began the process by first analyzing the problem, and then they determine the development criteria for the problem with some orientation. After that, they start suggesting a solution for the problem. Finally, they evaluate the suggested solution. It can be noticed that some students, e.g., Learner (1) did not collaborate with other group members. However, this learner should be more social with other group members. As a result, the social agent encourages Learner (1) for more collaboration. In addition, the teacher can evaluate the member interactions and send some hints to group members for problem clarification.

**Table 12.** A case study of FCS interactions sequence for Group 1.

| Learner Number | Interactions Sequence |
|---|---|
| Learner (1) | PA–PA |
| Learner (2) | PA–PC–OO |
| Learner (3) | NT–PA–CD–SD |
| Learner (4) | NT–NT–SE |
| Learner (5) | PA–PA–CD–NT–OO–SD–SD–SE |
| Learner (6) | PA–PA–CD–SE–NT |
| Learner (7) | PA–CD–SE–NT |
| Learner (8) | PA–PC–CD–SD–SD |

After building the interaction sequence, the social agent builds the FCS learner's participations ranking as shown in Table 13. It can be noticed that the group performance for class PA equals 10 and the total group performance is 28.

**Table 13.** A case study of FCS learners' participation ranking for Group 1.

| Class Name | Rank | Interactions Number | Total Rank |
|---|---|---|---|
| PA | 1 | 10 | 10 |
| PC | 1 | 2 | 2 |
| OO | 1 | 2 | 2 |
| CD | 1 | 5 | 5 |
| SD | 1 | 5 | 5 |
| SE | 1 | 4 | 4 |
| NT | 0 | 6 | 0 |
| **Total** | | | 28 |

Table 14 shows the FCS learner participation for Group 1. The record for learner participation within each class is presented in this table.

Table 15 shows the FCS learner's participation ratio for Group 1. The collaboration percentage for each learner within each class is presented in this table.

Figure 22 shows comparisons among learners' collaboration for each class of Group 1. It is obvious that Learner (5) has the most group participation in problem solving within the group (PA = 20%, OO = 50%, CD = 20%, SD = 40%, SE = 25%). On the other hand, Learner (1) has the minimum group participation (PA = 20%).

**Table 14.** A case study of FCS learner participation for Group 1.

| Learner/Class | PA | PC | OO | CD | SD | SE | NT |
|---|---|---|---|---|---|---|---|
| Learner (1) | 2 | 0 | 0 | 0 | 0 | 0 | 0 |
| Learner (2) | 1 | 1 | 1 | 0 | 0 | 0 | 0 |
| Learner (3) | 1 | 0 | 0 | 1 | 1 | 0 | 1 |
| Learner (4) | 0 | 0 | 0 | 0 | 0 | 1 | 2 |
| Learner (5) | 2 | 0 | 1 | 1 | 2 | 1 | 1 |
| Learner (6) | 2 | 0 | 0 | 1 | 0 | 1 | 1 |
| Learner (7) | 1 | 0 | 0 | 1 | 0 | 1 | 1 |
| Learner (8) | 1 | 1 | 0 | 1 | 2 | 0 | 0 |
| TOTAL | 10 | 2 | 2 | 5 | 5 | 4 | 6 |

**Table 15.** A case study of FCS learners' participation ratio.

| Learner/Class–Group 1 | PA% | PC% | OO% | CD% | SD% | SE% | NT% |
|---|---|---|---|---|---|---|---|
| Learner (1) | 20% | 0 | 0 | 0 | 0 | 0 | 0 |
| Learner (2) | 10% | 50% | 50% | 0 | 0 | 0 | 0 |
| Learner (3) | 10% | 0 | 0 | 20% | 20% | 0 | 16% |
| Learner (4) | 0 | 0 | 0 | 0 | 0 | 25% | 33% |
| Learner (5) | 20% | 0 | 50% | 20% | 40% | 25% | 16% |
| Learner (6) | 20% | 0 | 0 | 20% | 0 | 25% | 16% |
| Learner (7) | 10% | 50% | 0 | 20% | 0 | 25% | 16% |
| Learner (8) | 10% | 0 | 0 | 20% | 40% | 0 | 0 |

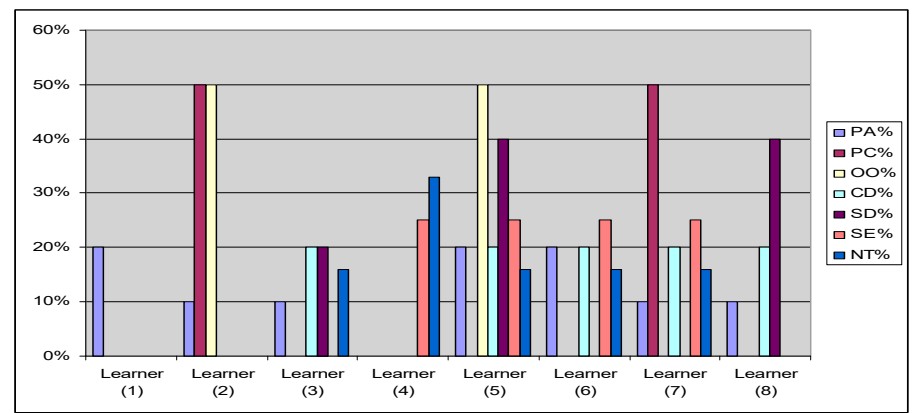

**Figure 22.** Learners' participation ratio graph for Group 1.

Similarly, learners' participation ratios for groups 2, 3, and 4 are shown in Figures 23–25, respectively.

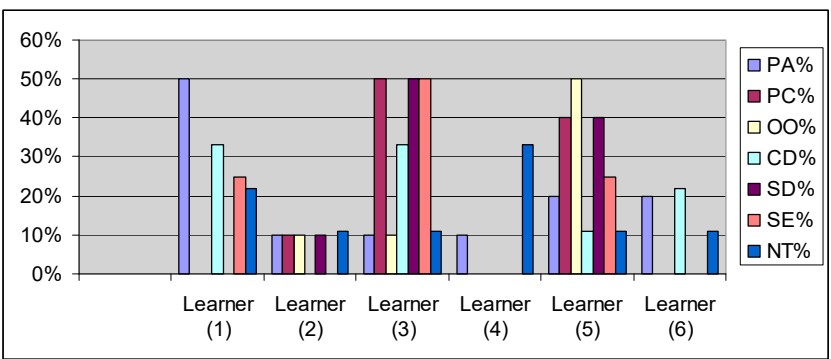

**Figure 23.** Learners' participation ratio for Group 2.

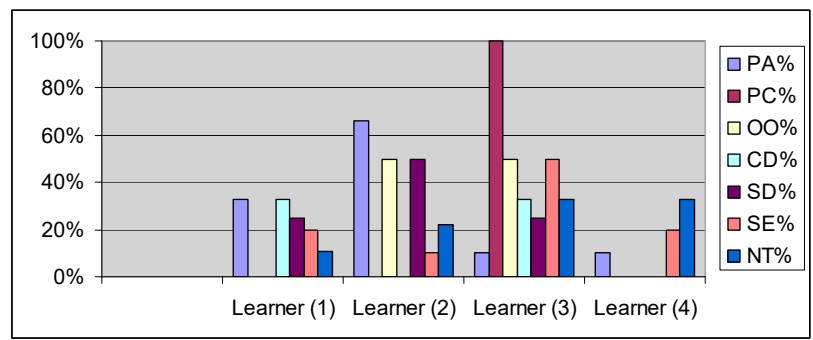

**Figure 24.** Learners' participation ratio for Group 3.

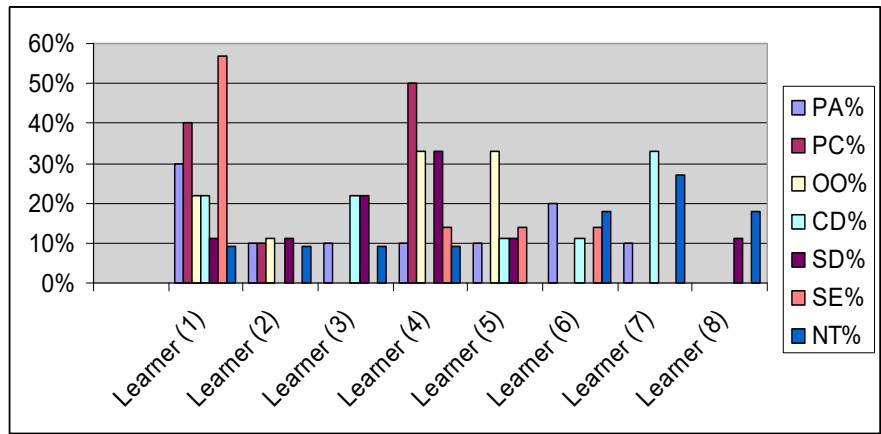

**Figure 25.** Learners' participation ratio for Group 4.

The comparison among the learners' collaboration within each class of Group 2 is shown in Figure 23. It is clear that Learner (3) (PA = 10%, PC = 50%, OO = 10%, CD = 33 %, SD = 50%, SE = 50%, NT = 11%) and Learner (5) (PA = 20%, PC = 40%, OO = 50%, CD = 11%, SD = 40%, SE = 25%, NT= 11%) have the most group participation in problem solving within the group. On the other hand, Learner (4) has the minimum group participation (Non-task = 33%, PA = 10%).

The comparison among the learners' collaboration within each class of Group 3 is shown in Figure 24. It is clear that Learner (3) has the most group participation in problem solving within the group (PA = 10%, PC = 100%, OO = 50%, CD = 33%, SD = 25%, SE = 50%, NT = 33%). On the other hand, Learner (4) has the minimum group participation (PA = 10%, SE = 20%, NT = 33%). Moreover, it can be noticed that Leaner (3) has PC = 100%. In this case, the social agent engages other leaners in the group to do more contributions with Learner (3) in class PC.

The comparison for learners' collaboration between each class of Group 4 is shown in Figure 25. It is clear that Learner (1) has the most group participation in problem solving within the group (PA = 30%, PC = 40%, OO = 22%, CD = 22%, SD = 11%, SE = 57%, NT = 9%). On the other hand, Learner (8) has the minimum group participation (SD = 10%, NT = 18%).

*7.3. Analyzing the Factors that Improved the Quality of Learning Process During the Experiment*

This section analyses the different factors that are used to improve the quality of learning process during this experiment in the proposed model. In this model, the social agent has the main role of assessing and evaluating member and group interactions, and it provides the following assessment to improve the collaboration during the learning process:

- Member quality of collaboration using FCS interactions sequence for collaborative groups.
- Group quality of collaboration using FCS learners' participations ranking factor for collaborative groups.

- Learners' performance using FCS learner's participation ratio.

For instance, the assessment of member's quality of collaboration parameters has direct correlation with FCS classes interactions sequence as shown in Figure 26, The members interact during problem solving as described in FCS classes sequence (PA–PC–OO–CD–SD–SE), therefore if an interaction is stopped at some point during problem solving, it will be noted and the situation will be assessed by the teacher.

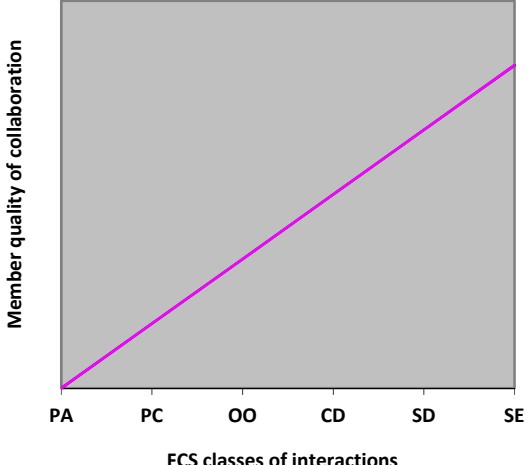

**Figure 26.** Member quality of collaboration assessment.

Additionally, the assessment of group quality of collaboration has direct correlation with FCS learners' participations ranking factor for collaborative groups as shown in Figure 27. If the group has a higher value ranking, it is more likely the group has more collaboration quality.

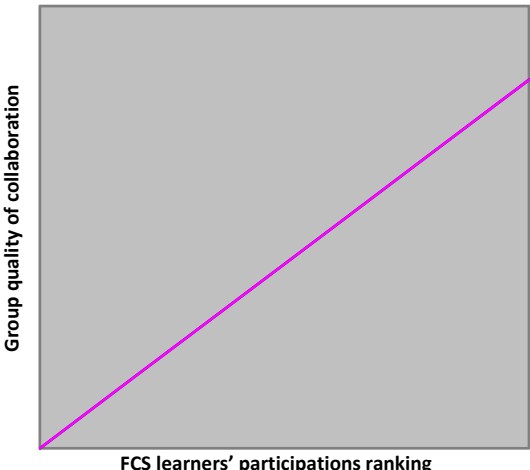

**Figure 27.** Group quality of collaboration assessment.

Furthermore, the assessment of the learners' performance has direct correlation with FCS learner's participation ratio as shown in Figure 28. If a learner has more participation ratio, this indicates that the learner has greater collaboration performance.

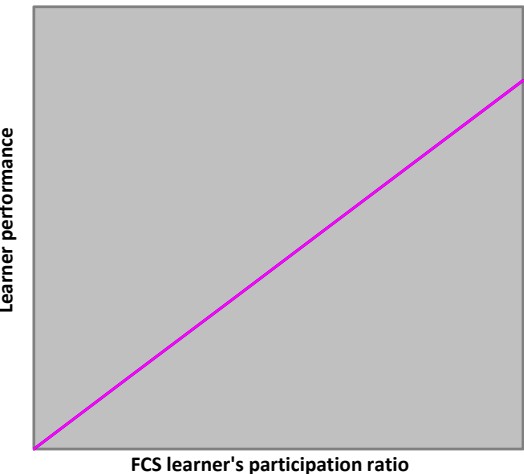

**Figure 28.** Group quality of collaboration assessment.

Based on the conducted experiments, the authors found that the proposed model can improve quality of learning process by assessing learners' and groups' collaboration, and it can help teachers make learners improve how they work in groups. Furthermore, it provides various ways of assessing learners' abilities and skills in groups.

## 8. Conclusions and Future Work

This work created an efficient collaborative m-learning model based on mobile agents and thus successfully implemented an m-learning architecture against the challenges addressed in this research, namely the support of efficient mobile learning in education and the complexity of incorporating technological and educational components together. The proposed model consists of four components: the learner agent, the teacher agent, the device agent and the social agent, where the social agent plays the main role through continuous evaluation and monitoring of the group collaboration. The proposed architecture model was implemented using the JADE-LEAP platform and was applied on mobile devices. The agent interactions were also analyzed to evaluate the performance of the proposed model. The questionnaire results showed that the proposed collaborative m-learning system was easy to use and access, well-organized, convenient, and facilitates the learning process. Additionally, the questionnaire's participants recommended using the proposed model to supplement classroom lectures. Moreover, the experimental results and simulation revealed that the proposed collaborative m-learning model enhanced the leaners' skills in problem solving, increased the learners' knowledge, and encouraged learners for more participation in the learning tasks.

The experiments and simulation of the proposed model also showed that the collaborative e-learning can be integrated with the collaborative m-learning. Several research directions are still needed to experiment with the performance, security, and compatibility issues between the two platforms. In addition, more investigation is needed to integrate the collaborative m-learning model based on agents with other e-learning and m-learning systems that have different architectural models. Furthermore, more research is needed to add other components to the proposed model to enhance the learning requirements, such as the teacher' profile that can be used by teachers to store their information.

**Author Contributions:** Conceptualization, S.A., M.A.-A., I.A., A.L. and M.A.; Methodology, M.A.-A., I.A. and A.L.; Software, I.A. and A.L.; Validation, M.A.-A., I.A. and A.L.; Formal Analysis, S.A. and A.L.; Investigation, M.A.; Resources, S.A., A.L. and M.A.; Data Curation, S.A., M.A.-A. and A.L.; Writing—Original Draft Preparation, S.A., M.A.-A., I.A. and A.L.; Writing—Review & Editing, S.A., M.A.-A., I.A. and M.A.; Supervision, M.A.A. and I.A.; Project Administration, S.A., M.A.-A. and I.A. All authors have read and agreed to the published version of the manuscript.

**Funding:** This research received no external funding and The APC was funded by the Deanship of Scientific Research at Saudi Electronic University.

**Conflicts of Interest:** The authors declare no conflict of interest.

## Appendix A. A Case Study Problem Description

In this appendix, the problem given to the students during the case study is discussed with more details about the conducted experiment.

### Appendix A.1. Problem Description Sent to Students Using the Teacher Agent

Suppose that x and y are stack objects. Explain why the overloaded assignment operator cannot be used in an expression x = y. Modify the prototype and implementation of the overloaded assignment operator so that this expression becomes valid.

### Appendix A.2. Hints Sent to Student during the Test

Please note the difference between Reference semantics and Value semantics: Reference Semantics means copying of reference (not the value in memory location; Value semantics means copying the value in memory location (without changing the reference in the memory.

Please note that copying overwrite pointer x.top_node, so the contents of x are lost and becomes garbage.

Also the two objects share the same nodes, so any destructor on x will result in the deletion of y.

Such a deletion would leave the pointer y.top_node addressing what has become a random memory location.

The problem caused by using the assignment operator on a linked stack arises because it copies a reference rather than a value.

### Appendix A.3. Functional Category System (FCS) Class Interaction Table

The FCS class interactions table used in test is shown in Table A1.

**Table A1.** FCS class interaction used in case study.

| Class Type | Interactions (KEY WORD) |
|---|---|
| Problem Analysis (PA) | think, analysis, understand, purpose, supposed, realize, recognize, reason, effect, show, illustrate, goal, aims, problem |
| Problem Critique (PC) | how, what, when, where, why |
| Orientation (OO) | find, go to, opinion, progress, check, wait, on track, look, need, include, draw, try, supply, next, pass |
| Criteria Development (CD) | parameters, variable, define, redefine, invoke, use, obtain, summarize, pointer, reference, value, semantics, stack, data, operator, test, prepare |
| Solution Development (SD) | solution, solve, suggest, idea, propose, task, code, function, make, clear, move, implement |
| Solution Evaluation (SE) | evaluate, agree, disagree, detail, good idea, bad idea, prove |
| Non-Task (NT) | break, sorry, hi, bye |

### Appendix A.4. Sample of Students' Interactions

**Mahmod**: I think that we will make the two parameters have the same index so any modification on any one of these two parameters will change the other.

**Dima**: I think the problem would become an object with no pointer which means that an object will be lost.

**Rawan**: We must understand the expression X = Y firstly.

**Dima**: I suggest assigning another pointer to the object that was connected to the variable x so that we do not lose any elements of the stack. What do you think? Do you agree with me?

**Saleh**: We need to define new object z, var's x,y,z.
**Dima:** I suggest that we take the value of y, insert t into z and modify the value of x
**Rawan**: I agree with Dima

**Appendix B. Collaborative M-learning Questionnaire**

This appendix lists the questions presented to the participants in the questionnaire. On a five-point scale (ranging from 1 = "strongly disagree" to 5 = "strongly agree"), students were asked to indicate their response to the following 21 statements about the collaborative m-learning system.

**Five-point scale**

(5) Strongly Agree     (4) Agree     (3) Neutral     (2) Disagree     (1) Strongly Disagree

**Questions**

1. The m-learning program was easy to use.
2. The m-learning program was well organized.
3. I feel the m-learning program facilitated learning.
4. The use of the m-learning program was convenient and easy to access.
5. The m-learning program was boring.
6. M-learning (mobile program) can never match the human contact that a teacher provides.
7. The m-learning program should be used in place of lectures.
8. The m-learning program should be used to supplement lectures.
9. Learning from m-learning is a cold and impersonal experience.
10. Learning from a mobile is an exciting way to learn.
11. I would rather learn from class lectures than m-learning program.
12. I like the combination of both lectures and m-learning.
13. I feel the m-learning program has helped me to develop my problem solving skills.
14. I feel that the m-learning exercise (mobile program) should be mandatory for all students in computer science.
15. I learned a lot from the mobile program.
16. My overall impression of the mobile program was that it was useful.
17. Overall, the mobile program was a valuable learning experience for me.
18. I enjoyed using the mobile program to enhance my problem solving skills.
19. I think the collaboration between students in m-learning is useful
20. The feedback received form social agent encouraged me for more participants in learning
21. The collaborative m-learning increases my knowledge rather than learning individually

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
