# Peer review of "Collaborative Mobile-Learning Architecture Based on Mobile Agents"

_electronics, doi:10.3390/electronics9010162_

Round 1

Reviewer 1 Report

This work presents a collaborative mobile learning architecture based on mobile agents. The paper is a bit too long, but easy to read, and information flows logically. Although not novel, the topic of the paper is interesting and relevant.

From my point of view, this study meets all the key requirements to be published. It includes a thorough literature review in which the background of the work is very well described. The agent-based model, architecture and implementation is clearly and profusely presented in several sections (this part is overly lengthy and could be summarised to facilitate reading). Finally, the authors carried out an empirical validation of their proposal in the form of a case study.

Reviewer 2 Report

The theme is very interesting and important in the era of IoT. 

I agree that m-learning has a great potential in improving the quality of learning process and has to to adopted by modern schools/universities, at least for distance learning. M- learning is important for the integration of modern technologies in the learning process, such as Virtual and Augmented Reality, Gamification etc.

I strongly disagree with the ideea that "in collaborative learning environment, students often learn better than in the standard classroom setting". The collaborative learning  is the most efficient method of teaching, and can be used in  standard classroom, too. It depends on teachers. Modern wise teachers use collaborative learning and IoT facilities in standard classroom with great succes.

I am not sure that this study is relevant, having in mind the very few numbers of students. Are they representative for the statistical population?  In which country/ region? In the graphs presented it can be seen that most of the students have medium activity. 

The authors could analyze which are the factors that improved the quality of learning process when using m-learning.

The  article seems more a literature review. 

I recommend the authors to read more WoS, SCOPUS, EBSCO, etc, articles published in the last 3 years.

Round 2

Reviewer 2 Report

The manuscript had been improved. 

Author Response

Responses to Reviewers Comments on the article entitled:

“Collaborative Mobile Learning Architecture based on Mobile Agents”

The authors would like to thank the editor and reviewer for their efforts and constructive comments:

The authors revised and proofread the manuscript as recommended by the reviewers.

Question:

The description of the research design, the methods and the results could still be more clearly presented. Novelty and the significance of the overall contribution should be better highlighted.

Response. Thank you for your comments. The above mentioned areas are clarified further and highlighted in the abstract, introduction and conclusion.

Sincere regards,

   Authors
